# A Differential Geometric View and Explainability of GNN on Evolving Graphs

**Yazheng Liu[†], Xi Zhang[†], Sihong Xie[*]**
[†]Key Laboratory of Trustworthy Distributed Computing and Service (MoE), BUPT, Beijing, China
[*]Department of Computer Science and Engineering, Lehigh University, Bethlehem, PA, USA
{liuyz,zhangx}@bupt.edu.cn, xiesihong1@gmail.com

## Abstract

Graphs are ubiquitous in social networks and biochemistry, where Graph Neural Networks (GNN) are the state-of-the-art models for prediction. Graphs can be evolving and it is vital to formally model and understand how a trained GNN responds to graph evolution. We propose a smooth parameterization of the GNN predicted distributions using axiomatic attribution, where the distributions are on a low-dimensional manifold within a high-dimensional embedding space. We exploit the differential geometric viewpoint to model distributional evolution as smooth curves on the manifold. We reparameterize families of curves on the manifold and design a convex optimization problem to find a unique curve that concisely approximates the distributional evolution for human interpretation. Extensive experiments on node classification, link prediction, and graph classification tasks with evolving graphs demonstrate the better sparsity, faithfulness, and intuitiveness of the proposed method over the state-of-the-art methods.

## 1 Introduction

Graph neural networks (GNN) are now the state-of-the-art method for graph representation in many applications, such as social network modeling Kipf & Welling (2017) and molecule property prediction Wu et al. (2018), pose estimation in computer vision Yang et al. (2021), smart cities Ye et al. (2020), fraud detection Wang et al. (2019), and recommendation systems Ying et al. (2018). A GNN outputs a probability distribution $\Pr(Y|G; \boldsymbol{\theta})$ of $Y$, the class random variable of a node (node classification), a link (link prediction), or a graph (graph classification), using trained parameters $\boldsymbol{\theta}$. Graphs can be evolving, with edges/nodes added and removed. For example, social networks are undergoing constant updates Xu et al. (2020a); graphs representing chemical compounds are constantly tweaked and tested during molecule design. In a sequence of graph snapshots, without loss of generality, let $G_0 \to G_1$ be *any* two snapshots where the source graph $G_0$ evolves to the destination graph $G_1$. $\Pr(Y|G_0; \boldsymbol{\theta})$ will evolve to $\Pr(Y|G_1; \boldsymbol{\theta})$ accordingly, and we aim to model and explain the evolution of $\Pr(Y|G; \boldsymbol{\theta})$ with respect to $G_0 \to G_1$ to help humans understand the evolution Ying et al. (2019); Schnake et al. (2020); Pope et al. (2019); Ren et al. (2021); Liu et al. (2021). For example, a GNN's prediction of whether a chemical compound is promising for a target disease during compound design can change as the compound is fine-tuned, and it is useful for the designers to understand how the GNN's prediction evolves with respect to compound perturbations.

To model graph evolution, existing work Leskovec et al. (2007; 2008) analyzed the macroscopic change in graph properties, such as graph diameter, density, and power law, but did not analyze how a parametric model responses to graph evolution. Recent work Kumar et al. (2019); Rossi et al. (2020); Kazemi et al. (2020); Xu et al. (2020b;a) investigated learning a model for each graph snapshot and thus the model is evolving, while we focus on modeling a fixed GNN model over evolving graphs. A more fundamental drawback of the above work is the discrete viewpoint of graph evolution, as individual edges and nodes are added or deleted. Such discrete modeling fails to describe the corresponding change in $\Pr(Y|G; \boldsymbol{\theta})$, which is generated by a computation graph that can be perturbed with infinitesimal amount and can be understood as a sufficiently smooth function. The smoothness can help identify subtle infinitesimal changes contributing significantly to change in $\Pr(Y|G; \boldsymbol{\theta})$, and thus more faithfully explain the change.

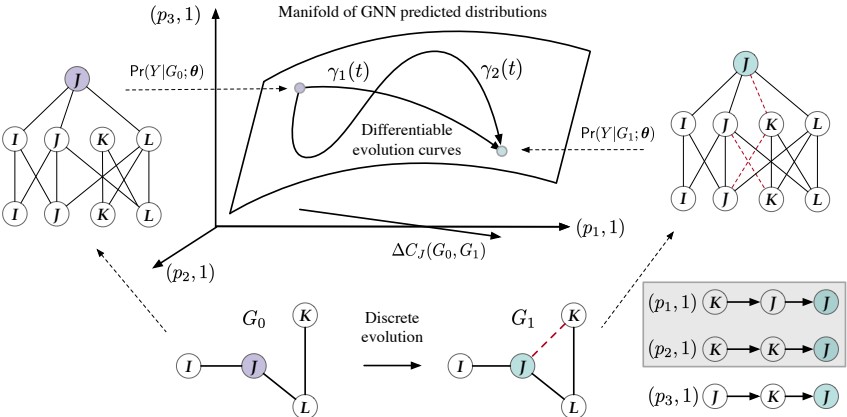

Figure 1: $G_0$ at time time $s = 0$ is updated to $G_1$ at time $s = 1$ after the edge $(J, K)$ is added, and the predicted class distribution $(\Pr(Y|G_0))$ of node $J$ changes accordingly. The contributions of each path $p$ on a computation graph to $\Pr(Y = j|G)$ for class $j$ give the coordinates of $\Pr(Y|G)$ in a high-dimensional Euclidean space, with axes indexed by $(p, j)$. $\Pr(Y|G)$ varies smoothly on a low dimensional manifold, where multiple curves $\gamma(s)$ can explain the evolution from $\Pr(Y|G_0)$ to $\Pr(Y|G_1)$ at very fine-grained. We select a $\gamma(s)$ that use a sparse set of axes for explaining the prediction evolution. Edge deletion, mixture of addition and deletion, link prediction, and graph classification are handled similarly.

Regarding explaining GNN predictions, there is promising progress made with static graphs, including local or global explanation methods Yuan et al. (2020b). Local methods explain individual GNN predictions by selecting salient subgraphs Ying et al. (2019), nodes, or edges Schnake et al. (2020). Global methods Yuan et al. (2020a); Vu & Thai (2020) optimize simpler surrogate models to approximate the target GNN and generate explaining models or graph instances. Existing counterfactual or perturbation-based methods Lucic et al. (2021) attribute a static prediction to individual edges or nodes by optimizing a perturbation to the input graph to maximally alter the target prediction, thus giving a sense of explaining graph evolution. However, the perturbed graph found by these algorithms can differ from $G_0$, and thus does not explain the change from $\Pr(Y|G_0; \boldsymbol{\theta})$ to $\Pr(Y|G_1; \boldsymbol{\theta})$. Both prior methods DeepLIFT Shrikumar et al. (2017) and GNN-LRP Schnake et al. (2020) can find propagation paths that contribute to prediction changes. However, they have a fixed $G_0$ for *any* $G_1$ and thus fail to model smooth evolution between arbitrary $G_0$ and $G_1$. They also handle multiple classes independently Schnake et al. (2020) or uses the log-odd of two predicted classes $Y = j$ and $Y = j'$ to measure the changes in $\Pr(Y|G; \boldsymbol{\theta})$ Shrikumar et al. (2017), rather than the overall divergence between two distributions.

To facilitate smooth evolution from $\Pr(Y|G_0; \boldsymbol{\theta})$ to $\Pr(Y|G_1; \boldsymbol{\theta})$ (we ignore the fixed $\boldsymbol{\theta}$ in the sequel), in Section 3.1, we set up a coordinate system for $\Pr(Y|G)$ using the contributions of paths on the computation graphs to $\Pr(Y|G)$ relative to a global reference graph. We rewrite the distribution $\Pr(Y|G)$ for node classification, link prediction, and graph classification using these coordinates, so that $\Pr(Y|G)$ for any graph $G$ is embedded in this Euclidean space spanned by the paths. For classification, the distributions $\Pr(Y|G)$ have only $c - 1$ sufficient statistics and form an intrinsic low-dimensional manifold embedded in the constructed extrinsic embedding Euclidean space. In Section 3.2, we study the curvature of the manifold *local* to a particular $\Pr(Y|G)$. We derive the Fisher information matrix $I$ of $\Pr(Y|G)$ with respect to the path coordinates. As the KL-divergence between $\Pr(Y|G_0)$ and $\Pr(Y|G_1)$ that are sufficiently close can be approximated by a quadratic function with the Fisher information matrix, $\Pr(Y|G)$ does not necessarily evolve linearly in the extrinsic coordinates but adapts to the curved intrinsic geometry of the manifold around $\Pr(Y|G)$. Previous explanation methods Shrikumar et al. (2017); Schnake et al. (2020) taking a linear view point will not sufficiently model such curved geometry. Our results justify KL-divergence as a faithfulness metric of explanations adopted in the literature. The manifold allows a set of curves $\gamma(s)$ with the continuous time variable $s \in [0, 1]$ to model differentiable evolution between two distributions. With a novel reparameterization, we devise a convex optimization problem to optimally select a curve depending on a small number of extrinsic coordinates to approximate the evolution of $\Pr(Y|G_0)$ into $\Pr(Y|G_1)$ following the local manifold curvature. Empirically, in Section 4, we show that the proposed model and algorithm help select sparse and salient graph elements to con-

cisely and faithfully explain the GNN responses to evolving graphs on 8 graph datasets with node classification, link prediction, and graph classification, with edge additions and/or deletions.

## 2 PRELIMINARIES

**Graph neural networks**. For *node classification*, assume that we have a trained GNN of $T$ layers that predicts the class distribution of each node $J \in \mathcal{V}$ on a graph $G = (\mathcal{V}, \mathcal{E})$. Let $\mathcal{N}(J)$ be the neighbors of node $J$. On layer $t$, $t = 1, \ldots, T$ and for node $J$, GNN computes hidden vector $\mathbf{h}_J^{(t)}$ using messages sent from its neighbors:

$$\mathbf{z}_J^{(t)} = f_{\text{UPDATE}}^{(t)}(f_{\text{AGG}}^{(t)}(\mathbf{h}_J^{(t-1)}, \mathbf{h}_K^{(t-1)}, K \in \mathcal{N}(J))), \tag{1}$$

$$\mathbf{h}_J^{(t)} = \text{NonLinear}(\mathbf{z}_J^{(t)}), \tag{2}$$

where $f_{\text{AGG}}^{(t)}$ aggregates the messages from all neighbors and can be the element-wise sum, average, or maximum of the incoming messages. $f_{\text{UPDATE}}^{(t)}$ maps $f_{\text{AGG}}^{(t)}$ to $\mathbf{z}_J^{(t)}$, using $\mathbf{z}_J^{(t)} = \left\langle f_{\text{AGG}}^{(t)}, \boldsymbol{\theta}^{(t)} \right\rangle$ or a multi-layered perceptron with parameters $\boldsymbol{\theta}^{(t)}$. For layer $t \in \{1, \ldots, T-1\}$, ReLU is used as the NonLinear mapping, and we refer to the linear terms in the argument of NonLinear as "*logits*". At the input layer, $\mathbf{h}_J^{(0)}$ is the node feature vector $\mathbf{x}_J$. At layer $T$, the logits are $\mathbf{z}_J^{(T)} \triangleq \mathbf{z}_J(G)$, whose $j$-th element $z_j(G)$ denotes the logit of the class $j = 1, \ldots, c$. $\mathbf{z}_J(G)$ is mapped to the class distribution $\Pr(Y_J|G; \boldsymbol{\theta})$ through the softmax ($c > 2$) or sigmoid ($c = 2$) function, and $\arg\max_j z_j = \arg\max_j \Pr(Y = j|G; \boldsymbol{\theta})$ is the predicted class for $J$. For *link prediction*, we concatenate $\mathbf{z}_I^{(T)}$ and $\mathbf{z}_J^{(T)}$ as the input to a linear layer to obtain the logits:

$$\mathbf{z}_{IJ} = \left\langle \left[\mathbf{z}_I^{(T)}; \mathbf{z}_J^{(T)}\right], \boldsymbol{\theta} \right\rangle. \tag{3}$$

Since link prediction is a binary classification problem, $Z_{IJ}$ can be mapped to the probability that $(I, J)$ exists using the sigmoid function. For *graph classification*, the average pooling of $\mathbf{z}_J(G)$ of all nodes from $G$ can be used to obtain a single vector representation $\mathbf{z}(G)$ of $G$ for classification.

Since the GNN parameters $\boldsymbol{\theta}$ are fixed, we ignore $\boldsymbol{\theta}$ in $\Pr(Y|G; \boldsymbol{\theta})$ and use $\Pr(Y|G)$ to denote the predicted class distribution of $Y$, which is a general random variable of the class of a node, an edge, or a whole graph, depending on the tasks. Similarly, we use $\mathbf{z}^{(t)}$ and $\mathbf{z}$ to denote logits on layer $t$ and the last layer of GNN, respectively. For a uniform treatment, we consider the GNN as learning node representations $\mathbf{z}_J$, while the concatenation, pooling, sigmoid, and softmax at the last layer that generate $\Pr(Y|G)$ from the node representations are task-specific and separated from the GNN.

Table 1: Symbols and their meanings

| Symbols | Definitions and Descriptions |
|---|---|
| $J, \ldots, U, V, \ldots, K$ | Nodes in the graph |
| $j, \ldots, u, v, \ldots, k$ | Neurons of the corresponding nodes |
| $c$ | The number of classes |
| $G_0 \rightarrow G_1$ | Graph $G_0$ evolves to $G_1$ |
| $\mathbf{z}_J(G)$ | Logit vector $[\mathbf{z}_1(G), \ldots, \mathbf{z}_c(G)]$ of node $J$ |
| $\Delta\mathbf{z}_J(G_0, G_1)$ | $\Delta\mathbf{z}_J(G_0, G_1) = \mathbf{z}_J(G_1) - \mathbf{z}_J(G_0)$ |
| $\Pr(Y|G)$ | Distribution $[\Pr_1(G), \ldots, \Pr_c(G)]$ of class $Y$ |
| $W(G)$ | Paths on the computation graph of GNN |
| $W_J(G)$ | The subset of $W(G)$ that computes $\mathbf{z}_J(G)$ |
| $\Delta W_J(G_0, G_1)$ | Altered paths in $W_J(G_0)$ as $G_0 \rightarrow G_1$ |
| $C_{p,j}$ | Contribution of the $p$-th altered path to $\Delta z_j$ |

**Evolving graphs**. In a sequence of graph snapshots, let $G_0 = (\mathcal{V}_0, \mathcal{E}_0)$ denote an arbitrary source graph with edges $\mathcal{E}_0$ and nodes $\mathcal{V}_0$, and $G_1 = (\mathcal{V}_1, \mathcal{E}_1)$ an arbitrary destination graph, so that the edge set evolves from $\mathcal{E}_0$ to $\mathcal{E}_1$ and the node set evolves from $\mathcal{V}_0$ to $\mathcal{V}_1$. We denote the evolution by $G_0 \rightarrow G_1$. Both sets can undergo addition, deletion, or both, and all such operations happen deterministically so that the evolution is discrete. Let $\Delta\mathcal{E}$ be the set of altered edges: $\Delta\mathcal{E} = \{e : e \in \mathcal{E}_1 \wedge e \notin \mathcal{E}_0 \text{ or } e \in \mathcal{E}_0 \wedge e \notin \mathcal{E}_1\}$. As $G_0 \rightarrow G_1$, there is an evolution from $\Pr(Y|G_0)$ to $\Pr(Y|G_1)$.

**Definition** (*Differentiable evolution*): *Given a fixed GNN model, for $G_0 \rightarrow G_1$, find a family of computational models $\{\Pr(Y|G(s)) : s \in [0, 1], \Pr(Y|G(0)) = \Pr(Y|G_0), \Pr(Y|G(1)) = \Pr(Y|G_1)\}$ and $\Pr(Y|G(s))$ is differentiable with respect to the time variable $s$.* [1]

**Differential geometry**. An $n$ dimensional manifold $\mathcal{M}$ is a set of points, each of which can be associated with a local $n$ dimensional Euclidean tangent space. The manifold can be embedded in a

---

[1] A computational model that outputs $\Pr(Y|G(s))$ does not necessarily correspond to a concrete input graph $G(s)$. We use the notation $\Pr(Y|G(s))$ and $G(s)$ for notation convenience only.

global Euclidean space $\mathbb{R}^m$, so that each point can be assigned with $m$ global coordinates. A smooth curve on $\mathcal{M}$ is a smooth function $\gamma : [0, 1] \to \mathcal{M}$. A two dimensional manifold embedded in $\mathbb{R}^3$ with two curves is shown in Figure 1.

## 3 DIFFERENTIAL GEOMETRIC VIEW OF GNN ON EVOLVING GRAPHS

### 3.1 EMBED A MANIFOLD OF GNN PREDICTED DISTRIBUTIONS

While a manifold in general is coordinate-free, we aim to embed a manifold in an extrinsic Euclidean space for a novel parameterization of $\Pr(Y|G)$. The GNN parameter $\boldsymbol{\theta}$ is fixed and cannot be used. $\Pr(Y|G)$ is given by the softmax or sigmoid of the sufficient statistics $\mathbf{z}(G)$, which are used as coordinates in information geometry Amari (2016). However, the logits sit at the ending layer of the GNN and will not fully capture how graph evolution influences $\Pr(Y|G)$ through the changes in the computation of the logits. GNNExplainer Ying et al. (2019) adopts a soft element-wise mask over node features or edges and thus parameterize $\Pr(Y|G)$ using the mask. However, the mask works on the input graph (edges or node features), without revealing changes in the internal GNN computation process.

We propose a novel extrinsic coordinate based on the contributions of paths to $\Pr(Y|G)$ on the computation graph of the GNN. $\mathbf{z}_J(G)$ is generated by the computation graph of the given GNN, which is a spanning tree rooted at $J$ of depth $T$. Figure 1 shows two computation graphs for $G_0$ and $G_1$. On a computation graph, each node consists of neurons for the corresponding node in $G$, and we use the same labels $(I, J, K, L, \text{etc.})$ to identify nodes in the input and computational graphs. The leaves of the tree contain neurons from the input layer $(t = 0)$ and the root node contains neurons of the output layer $(t = T)$. The trees completely represent the computations in Eqs. (1)-(2), where each message is passed through a path from a leaf to the root. Let a path be $(\dots, U, V, \dots, J)$, where $U$ and $V$ represent any two adjacent nodes and $J$ is the root where $\mathbf{z}_J(G)$ is generated. For a GNN with $T$ layers, the paths are sequences of $T + 1$ nodes. Let $W_J(G)$ be the paths ending at $J$.

Consider a reference graph $G^*$ containing all nodes in the graphs during the evolution. The symmetric set difference $\Delta W_J(G^*, G) = W_J(G^*)\Delta W_J(G)$ contains all $m$ paths rooted at $J$ with at least one altered edge when $G^* \to G$. For example, in Figure 1, $\Delta W_J(G^*, G) = \{(J, K, J), (K, J, J), (K, K, J), (L, K, J)\}$. $\Delta W_J(G^*, G)$ causes the change in $\mathbf{z}_J$ and $\Pr(Y|G)$ through the computation graph. Let the difference in $\mathbf{z}_J$ computed on $G^*$ and $G$ be $\Delta \mathbf{z}_J(G^*, G) = \mathbf{z}_J(G) - \mathbf{z}_J(G^*) = [\Delta z_1, \dots, \Delta z_c] \in \mathbb{R}^c$. We adopt DeepLIFT to GNN (see Shrikumar et al. (2017) and Appendix A.4) to compute the contribution $C_{p,j}(G)$ of each path $p \in \Delta W_J(G^*, G)$ to $\Delta z_j(G)$ for any class $j = 1, \dots, c$, so that $[z_1(G), \dots, z_c(G)]$ is reparameterized as

$$[z_1(G^*), \dots, z_c(G^*)] + \left[\sum_{p=1}^{m} C_{p,1}(G), \dots, \sum_{p=1}^{m} C_{p,c}(G)\right] = \mathbf{z}_J(G^*) + \mathbf{1}^\top C_J(G) \qquad (4)$$

Here, $C_J(G)$ is the contribution matrix with $C_{p,j}(G)$ as elements and $C_{:j}(G)$ the $j$-th column. $\mathbf{1}$ is an all-1 $m \times 1$ vector. By fixing $G^*$ and $\mathbf{z}_J(G^*)$, we use $C_{p,j}(G)$ as the extrinsic coordinates of $\Pr(Y|G)$. In this coordinates system, the difference vector between two logits for node $J$ is:

$$\Delta \mathbf{z}_J(G_0, G_1) = \mathbf{z}_J(G_1) - \mathbf{z}_J(G_0) = \mathbf{1}^\top(C_J(G_1) - C_J(G_0)) = \mathbf{1}^\top \Delta C_J(G_0, G_1). \qquad (5)$$

If we set $G^* = G_0$, we have $\Delta \mathbf{z}_J(G_0, G_1) = \mathbf{1}^\top C_J(G_1)$. Even with a fixed $G^*$, different graphs $G$ and nodes $J$ can lead to different sets of $\Delta W_J(G^*, G)$. We obtain a unified coordinate system by taking the union $\cup_{G,J \in G}\Delta W_J(G^*, G)$. We set the rows of $C_J(G)$ to zeros for those paths that are not in $\Delta W_J(G^*, G)$. In implementing our algorithm, we only rely on the observed graphs to exhaust the relevant paths without computing $\cup_{G,J \in G}\Delta W_J(G^*, G)$. We now embed $\Pr(Y|G)$ in the coordinate system.

- Node classification: the class distribution of node $J$ is
$$\Pr(Y|G) = \text{softmax}(\mathbf{z}_J(G)) = \text{softmax}(\mathbf{z}_J(G^*) + \mathbf{1}^\top C_J(G)). \qquad (6)$$

- Link prediction: for a link $(I, J)$ between nodes $I$ and $J$, the logits $\mathbf{z}_I(G)$ and $\mathbf{z}_J(G)$ are concatenated as input to a linear layer $\boldsymbol{\theta}_{\text{LP}}$ ("LP" means "link prediction"). The class distribution of the link is
$$\Pr(Y|G) = \text{sigmoid}([\mathbf{z}_I(G^*) + \mathbf{1}^\top C_I(G); \mathbf{z}_J(G^*) + \mathbf{1}^\top C_J(G)]\boldsymbol{\theta}_{\text{LP}}). \qquad (7)$$

- Graph classification: with a linear layer $\boldsymbol{\theta}_{\mathrm{GC}}$ ("GC" for "graph classification") and average pooling, the distribution of the graph class is

$$\mathsf{Pr}(Y|G) = \mathrm{softmax}(\mathrm{mean}(\mathbf{z}_J(G^*) + \mathbf{1}^\top C_J(G) : J \in \mathcal{V})\boldsymbol{\theta}_{\mathrm{GC}}). \tag{8}$$

The arguments of the above softmax and sigmoid are linear in $C_J(G)$ for all $J \in \mathcal{V}$ of $G$, and we recover exponential families reparameterized by $C_J(G)$. For a specific prediction task, we let the contribution matrices $C_J(G)$ in the corresponding equation of Eqs. (6)-(8) vary smoothly, and the resulting set $\{\mathsf{Pr}(Y|G)\}$ constitutes a manifold $\mathcal{M}(G, J)$. The dimension of the manifold is the same as the number of sufficient statistics of $\mathsf{Pr}(Y|G)$, though the embedding Euclidean space has $mc$ ($2mc$ and $|\mathcal{V}|mc$, resp.) coordinates for node classification (link prediction and graph classification, resp.), where $m$ is the number of paths in $\Delta W_J(G_0, G_1)$ and $c$ the number of classes.

## 3.2 A CURVED METRIC ON THE MANIFOLD

We will define a curved metric on the manifold $\mathcal{M}(G, J)$ of node classification probability distribution (link prediction and graph classification can be done similarly). A well-defined metric is vital to tasks such as metric learning on manifolds, which we will use to explain evolving GNN predictions in Section 3.3. We could have approximated the distance between two distributions $\mathsf{Pr}(Y|G_0)$ and $\mathsf{Pr}(Y|G_1)$ by $\|\Delta C_J(G_0, G_1)\|$ with some matrix norm (e.g., the Frobenius norm), as shown in Figure 1. As another example, DeepLIFT Shrikumar et al. (2017) uses the linear term $\mathbf{1}^\top(C_{:j}(G_0) - C_{:j'}(G_1))$ for two predicted classes $j$ and $j'$ on $G_0$ and $G_1$, respectively. These options imply the Euclidean distance metric defined in the flat space spanned by elements in $C_J(G)$. However, the evolution of $\mathsf{Pr}(Y|G)$ on $\mathcal{M}(G, J)$ depends on $C_J(G_0)$ and $C_J(G_1)$ nonlinearly through the sigmoid or softmax function as in Eqs. (6)-(8), and the difference between $\mathsf{Pr}(Y|G_0)$ and $\mathsf{Pr}(Y|G_1)$ should reflect the curvatures over the manifold $\mathcal{M}(G, J)$ of class distributions.

We adopt information geometry Amari (2016) to defined a curved metric on the manifold $\mathcal{M}(G, J)$. Take node classification as an example, the KL-divergence $D_{\mathrm{KL}}(\mathsf{Pr}(Y|G_1)\|\mathsf{Pr}(Y|G_0))$ between any two class distributions on the manifold $\mathcal{M}(G, J)$ is defined as $\sum_Y \log \mathsf{Pr}(Y|G_1)\frac{\mathsf{Pr}(Y|G_1)}{\mathsf{Pr}(Y|G_0)}$. As the parameter $C_J(G_0)$ approaches $C_J(G_1)$, $\mathsf{Pr}(Y|G_0)$ becomes close to $\mathsf{Pr}(Y|G_1)$ (as measured by the following Riemannian metric on $\mathcal{M}(G, J)$, rather than the Euclidean metric of the extrinsic space), and the KL-divergence can be approximated locally at $\mathsf{Pr}(Y|G_1)$ as

$$\mathbf{vec}(\Delta C_J(G_1, G_0))^\top I(\mathbf{vec}(C_J(G_1)))\mathbf{vec}(\Delta C_J(G_1, G_0)), \tag{9}$$

where $\mathbf{vec}(C_J(G_1))$ is the column vector with all elements from $C_J(G_1)$, and similar for $\mathbf{vec}(\Delta C_J(G_1, G_0))$ with the matrix $\Delta C_J(G_1, G_0)$. $I(\mathbf{vec}(C_J(G_1)))$ is the Fisher information matrix of the distribution $\mathsf{Pr}(Y|G_1)$ with respect to parameters in $\mathbf{vec}(C_J(G_1))$, evaluated as $(\nabla_{\mathbf{vec}(C_J(G_1))}\mathbf{z}_J(G_1))^\top \mathbb{E}_{Y \sim \mathsf{Pr}(Y|G_1)}[s_{\mathbf{z}_J(G_1)}s_{\mathbf{z}_J(G_1)}^\top](\nabla_{\mathbf{vec}(C_J(G_1))}\mathbf{z}_J(G_1))$, with $s_{\mathbf{z}_J(G_1)} = \nabla_{\mathbf{z}_J(G_1)}\log \mathsf{Pr}(Y|G_1)$ being the gradient vector of the log-likelihood with respect to $\mathbf{z}_J(G_1)$ and $\nabla_{\mathbf{vec}(C_J(G_1))}\mathbf{z}_J(G_1) \in \mathbb{R}^{c \times (mc)}$ the Jacobian of $\mathbf{z}_J(G_1)$ w.r.t $\mathbf{vec}(C_J(G_1))$. See (Martens (2020) and Appendix A.2 for the derivations). $I$ is symmetric positive definite (SPD) and Eq. (9) defines a non-Euclidean metric on the manifold to make $\mathcal{M}$ a Riemannian manifold.

## 3.3 CONNECTING TWO DISTRIBUTIONS VIA A SIMPLE CURVE

We formulate the problem of explaining the GNN prediction evolution as optimizing a curve on the manifold $\mathcal{M}$. Take node classification as an example [2]. Let $s \in [0, 1]$ be the time variable. As $s \to 1$, $\mathsf{Pr}(Y|G(s))$ moves smoothly over $\mathcal{M}$ along a curve $\gamma(s) \in \Gamma(G_0, G_1) = \{\gamma(s) = \{\mathsf{Pr}(Y|G(s)) : s \in [0, 1], \mathsf{Pr}(Y|G(0)) = \mathsf{Pr}(Y|G_0), \mathsf{Pr}(Y|G(1)) = \mathsf{Pr}(Y|G_1)\} \subset \mathcal{M}\}$. Two possible curves $\gamma_1(s)$ and $\gamma_2(s)$ are shown in Figure 1. With the parameterization in Eqs. (5)-(6), we can specifically define the following curves by smoothly varying the path contributions to $\mathsf{Pr}(Y|G(s))$ through $\Delta\mathbf{z}_J(G(s))$ ($s$ can be reversed to move in the opposite direction along $\gamma(s)$).

- linear in the directional matrix $\Delta C_J(G_0, G_1)$, with $\Delta C_J(G_0, G(s)) = \Delta C_J(G_0, G_1)s$;
- linear in the elements of $\Delta C_J(G_0, G_1)$: $\Delta C_J(G_0, G(s)) = \Delta C_J(G_0, G_1) \odot X(s)$, where $\odot$ is element-wise product and the matrix element $X(s)_{p,j}$ is a function mapping $s \in [0, 1] \to [0, 1]$;

---

[2] In the Appendix section A.3, we discuss the cases of link prediction and graph classification.

- linear in the rows of $\Delta C_J(G_0, G_1)$: let $\vec{\mathbf{x}} = [x_1(s), \ldots, x_m(s)]^\top$ and $x_p(s) \in [0, 1]$ weight the $p$-th path as a whole, and

$$\Delta C_J(G_0, G(s)) = \Delta C_J(G_0, G_1) \odot [\mathbf{1}_{1 \times c} \otimes \vec{\mathbf{x}}(s)], \tag{10}$$

where $\otimes$ is the Kronecker product creating the path weighting matrix $\mathbf{1}_{1 \times c} \otimes \vec{\mathbf{x}}(s) \in [0, 1]^{mc}$.

According to the derivation in Appendix A.1, we can rewrite $D_{\mathrm{KL}}(\mathsf{Pr}(Y|G_1)||\mathsf{Pr}(Y|G_0))$ as

$$\mathbb{E}_{j \sim \mathsf{Pr}(Y|G_1)}[\mathbf{1}^\top (C_{:j}(G_1) - C_{:j}(G_0))] - \log Z(G_1) + \log \sum_{j=1}^c \exp\{z_j(G^*) + \mathbf{1}^\top C_{:j}(G_0)\} \tag{11}$$

where the expectation has class $j$ sampled from $\mathsf{Pr}(Y|G_1)$ and $\log Z(G_1)$ is the cumulant function of $\mathsf{Pr}(Y|G_1)$. In Eq. (11), by letting $G_0$ vary along any $\gamma(s)$ as parameterized above and replacing $C_{:j}(G_0)$ with $C_{:j}(G(s)) = C_{:j}(G_0) + \Delta C_{:j}(G_0, G(s))$, we obtain $D_{\mathrm{KL}}(\mathsf{Pr}(Y|G_1)||\mathsf{Pr}(Y|G(s)))$. Taking $s \to 1$, the curve $\mathsf{Pr}(Y|G(s))$ enters a neighborhood of $\mathsf{Pr}(Y|G_1)$ on the manifold $\mathcal{M}$ to approximate $\mathsf{Pr}(Y|G_1)$ and $D_{\mathrm{KL}}(\mathsf{Pr}(Y|G_1)||\mathsf{Pr}(Y|G(s))) \to 0$ so that the curve $\gamma(s)$ parameterized by $\vec{\mathbf{x}}(s)$ smoothly mimics the movement from $\mathsf{Pr}(Y|G_0)$ to $\mathsf{Pr}(Y|G_1)$, at least locally in the neighborhood of $\mathsf{Pr}(Y|G_1)$. Since $\gamma(s) \in \Gamma(G_0, G_1) \subset \mathcal{M}(G, J)$, selecting a curve $\gamma(s)$ is different from selecting some edges from $G_1$ to approximate the distribution $\mathsf{Pr}(Y|G_1)$ as in Ying et al. (2019). Rather, the curves should move according to the geometry of the manifold $\mathcal{M}(G, J)$.

We can use Eq. (11) to explain how the computation of $\mathsf{Pr}(Y|G_0)$ evolves to that of $\mathsf{Pr}(Y|G_1)$ following $\gamma(s)$. There are $mc$ coordinates in $C_J(G(s))$, and can be large when $J$ is a high-degree node, while an explanation should be concise. We will identify a curve $\gamma(s)$ using a small number of coordinates for conciseness. The parameterization Eq. (10) assigns a weight $x_p(s)$ to each path $p$ at time $s$, allowing thresholding the elements in $\vec{\mathbf{x}}(s)$ to select a subset $E_n$ of $n \ll m$ paths. The contributions from these few selected path is now $\Delta C_J(G_0, G(s))$, which should well-approximate $\Delta C_J(G_0, G(1))$ as $s \to 1$. For example, in Figure 1, we can take $E_2 = \{(K, J, J), (L, K, J)\} \subset \Delta W_J(G_0, G_1)$ with $n = 2$. The selected paths span a low-dimensional space to embed the neighborhood of $\mathsf{Pr}(Y|G_1)$ on the manifold $\mathcal{M}(J, G)$. Adding paths in $E_n$ to the computation graph of $G_0$ leads to a new computation graph on the manifold.

We optimize $E_n$ to minimize the KL-divergence in Eq. (11) with Eq. (10). Let $x(s; p) \in [0, 1]$, $p = 1, \ldots, m$ be the weight of selecting path $p$ into $E_n$. We solve the following problem:

$$\min_{\substack{\vec{\mathbf{x}}(s) \in [0,1]^m \\ \|\vec{\mathbf{x}}(s)\|_1 = n}} \mathbb{E}_{j \sim \mathsf{Pr}(Y=j|G_1)}[\mathbf{1}^\top (C_{:j}(G_1) - C_{:j}(\vec{\mathbf{x}}(s)))] + \log \sum_{j=1}^c \exp\{z_j(G^*) + \mathbf{1}^\top C_{:j}(\vec{\mathbf{x}}(s))\} \tag{12}$$

where $C_{:j}(\vec{\mathbf{x}}(s)) = C_{:j}(G_0) + \Delta C_{:j}(G_0, G(s))$ is a vector of path contributions to the logit of class $j$. $\Delta C_{:j}(G_0, G(s))$ is parameterized by Eq. (10) and is a function of $\vec{\mathbf{x}}(s)$. The constants $\log Z(G_1)$ is ignored from Eq. (11) as $G_1$ is fixed. The linear constraint ensures the total probabilities of the selected edges is $n$. The optimization problem is convex and has a unique optimal solution. We select the paths with the highest $\vec{\mathbf{x}}(s)$ values to constitute a curve $\gamma(s)$ that explains the change from $\mathsf{Pr}(Y|G_0)$ to $\mathsf{Pr}(Y|G_1)$ as $\gamma(s)$ approaches $\mathsf{Pr}(Y|G_1)$. Concerning the Riemannian metric in Eq. (9), the above optimization does not change the Riemannian metric $I(\mathbf{vec}(C_J(G_1)))$ at $\mathsf{Pr}(Y|G_1)$ since the objective function is based on the KL-divergence of distributions generated by the non-linear softmax mapping, while $C_{:j}(\vec{\mathbf{x}}(s))$ vary in the extrinsic coordinate system with $\vec{\mathbf{x}}(s)$.

## 4 EXPERIMENTS

**Datasets and tasks**. We study node classification task on evolving graphs on the YelpChi, Yelp-NYC, YelpZip Rayana & Akoglu (2015), Pheme Zubiaga et al. (2017) and Weibo Ma et al. (2018) datasets, and study the link prediction task on the BC-OTC, BC-Alpha, and UCI datasets. These datasets have time stamps and the graph evolutions can be identified. The molecular data (MUTAG Debnath et al. (1991) is used for the graph classification. In searching molecules, slight perturbations are applied to molecule graphs You et al. (2018). We simulate the perturbations by randomly add or remove edges to create evolving graphs. Appendix A.5.1 gives more details.

**Experimental setup**. For each dataset, we optimize a GNN parameter $\boldsymbol{\theta}$ on the training set of static graphs, using labeled nodes, edges, or graphs, depending on the tasks. For each graph snapshot except the first one, target nodes/edges/graphs with a significantly large $D_{\mathrm{KL}}(\mathsf{Pr}(Y|G_0)||\mathsf{Pr}(Y|G_1))$

are collected and the change in $\Pr(Y|G)$ is explained. We run Algorithm 1 to calculate the contribution matrix $C_J(G)$ for each node $J \in \mathcal{V}^*$. We use the cvxpy library Diamond & Boyd to solve the constrained convex optimization problem in Eq. (12), Eq.(14) and Eq.(15). This method is called "**AxiomPath-Convex**". We also adopt the following baselines.

- **Gradient**. Grad computes the gradients of the logit of the predicted class $j$ with the maximal $\Pr(Y = j|G)$ on $G_0$ and $G_1$, respectively. Each computation path is assigned the sum of gradients of the edges on the paths as its importance. The contribution of a path to the change in $\Pr(Y|G)$ is the difference between the two path importance scores computed on $G_0$ and $G_1$. If a path only exists on one graph, the importance of the path is taken as the contribution. All paths with top importance are selected into $E_n$.

- **GNNExplainer** (GNNExp) Ying et al. (2019) is designed to explain GNN predictions for node and graph classification on static graphs. It weight edges on $G_1$ to maximally preserve $\Pr(Y|G_1)$ regardless of $\Pr(Y|G_0)$. Paths are weighted and selected as for Grad, with edge weights calculated using GNNExplainer.

- **GNN-LRP** adopts the back-propagation attribution method LRP to GNN Schnake et al. (2020). It attributes the class probability $\Pr(Y = j|G_1)$ to input neurons regardless of $\Pr(Y|G_0)$. It assigns an importance score to paths and top paths are put in $E_n$.

- **DeepLIFT** Shrikumar et al. (2017) can attribute the log-odd between two probabilities $\Pr(Y = j|G_0)$ and $\Pr(Y = j'|G_1)$, where $j \neq j'$. For a target node or edge or graph, if the predicted class changes, the difference between a path's contributions to the new and original predicted classes is used to rank and select paths. If the predicted class remains the same but the distribution changes, a path's contributions to the same predicted class is used. Only paths from $\Delta W_J(G_0, G_1)$ or in $\Delta W_I(G_0, G_1) \cup \Delta W_J(G_0, G_1)$ or in $\cup_{J \in \mathcal{V}} \Delta W_J(G_0, G_1)$ are ranked and selected.

- **AxiomPath-Topk** is a variant of AxiomPath-Convex. It selects the top paths $p$ from $\Delta W_J(G_0, G_1)$ or $\Delta W_I(G_0, G_1) \cup \Delta W_J(G_0, G_1)$ or $\cup_{J \in \mathcal{V}} \Delta W_J(G_0, G_1)$ with the highest contributions $\Delta C_J(G_0, G_1)\mathbf{1}$, where $\mathbf{1}$ is an all-1 $c \times 1$ vector. This baseline works in the Euclidean space spanned by the paths as coordinates and rely on linear differences in $C(G)$ rather than the nonlinear movement from $\Pr(Y|G_0)$ to $\Pr(Y|G_1)$.

- **AxiomPath-Linear** optimizes the AxiomPath-Convex objectives without the last log terms, leading to a linear programming.

**Quantitative evaluation metrics**. Let $\Pr(Y|\neg G(s))$ be computed on the computation graph of $G_1$ with those from $E_n$ disabled. That should bring $G_1$ close to $G_0$ along $\gamma(s)$ so that $\mathbf{KL}^+ = D_{\mathrm{KL}}\Pr_J(G_0)\Pr_J(\neg G(s))$ should be small if $E_n$ does contain the paths vital to the evolution. Similarly, we expect $\Pr_J(G(s))$ to move close to $\Pr_J(G_1)$ after the paths $E_n$ are enabled on the computation graph of $G_0$, and $\mathbf{KL}^- = \mathrm{KL}(\Pr_J(G_1)\|\Pr_J(G(s)))$ should be smaller. Intuitively, if $E_n$ indeed contains the more salient altered paths that turn $G_0$ into $G_1$, the less information the remaining paths can propagate, the more similar should $G_n$ be to $G_1$ and $\neg G_n$ be to $G_0$, and thus the *smaller* the KL-divergence. Prior work Suermondt (1992); Yuan et al. (2020a); Ying et al. (2019) use KL-divergence to measure the approximation quality of a static predicted distribution $\Pr(Y|G)$, while the above metrics evaluate how distribution on the curve $\gamma(s)$ approach the target $\Pr(Y|G_1)$. A similar metric can be defined for the link prediction task and the graph classification task, where the KL-divergence is calculated using predicted distributions over the target edge or graph. The target nodes (links or graphs ) are grouped based on the number altered paths in $\Delta W_J(G_0, G_1)$ for the results to be comparable, since alternating different number of paths can lead to significantly different performance. For each group, we let $n = |E_n|$ range in a pre-defined 5-level of explanation simplicity and all methods are compared under the same level of simplicity. Appendix A.5.2 and Appendix A.5.3 gives more details of the experimental setup.

## 4.1 PERFORMANCE EVALUATION AND COMPARISON

We compare the performance of the methods on three tasks (node classification, link prediction and graph classification) under different graph evolutions (adding and/or deleting edges). For the node classification, in Figure 2, we demonstrate the effectiveness of the salient path selection of AxiomPath-Convex. For each dataset, we report the average $\mathrm{KL}^+$ over target nodes/edges on three datasets (results with the $\mathrm{KL}^-$ metric, and results on the remaining datasets are given Figure 6, 7,

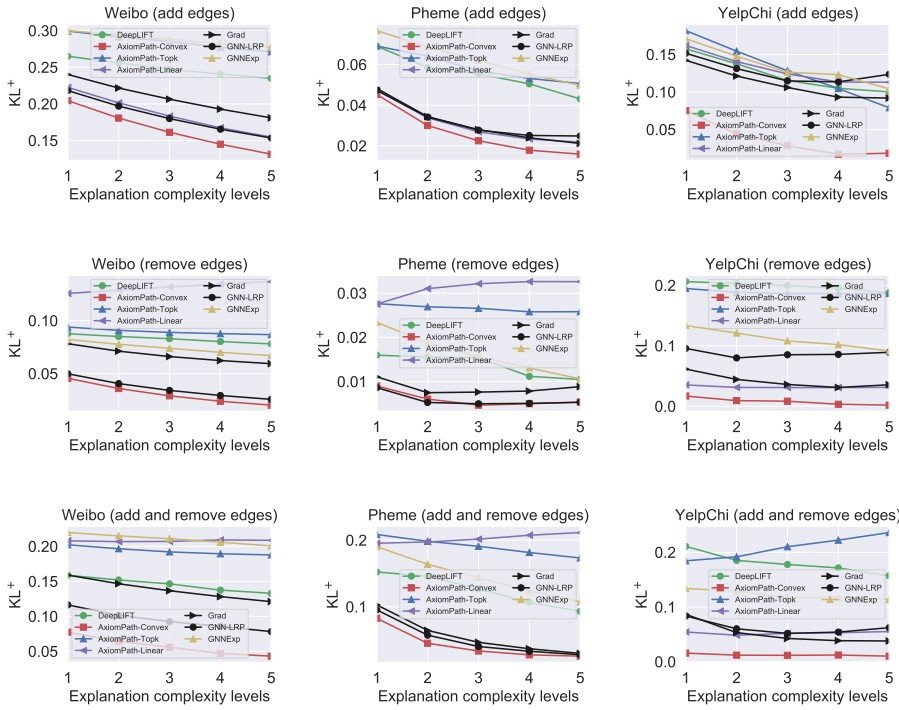

Figure 2: Performance in $KL^+$ as $G_0 \to G_1$ on the node classification tasks. Each column is a dataset and each row is one type of evolution.

and 8 in the Appendix). From the figures, we can see that AxiomPath-Convex has the smallest $KL^+$ over all levels of explanation complexities and over all datasets. On six settings (Weibo-adding edges only and mixture of adding and removing edges, and all cases on YelpChi), the gap between AxiomPath-Convex and the runner-up is significant. On the remaining settings, AxiomPath-Convex slightly outperforms or is comparable to the runner-ups. AxiomPath-Topk and AxiomPath-Linear underperform AxiomPath-Convex, indicating that modeling the geometry of the manifold of probability distributions has obvious advantage over working in the linear parameters of the distributions. On two link prediction tasks and one graph classification task, in Figure 3, we show that AxiomPath-Convex significantly uniformly outperform the runner-ups (results on the remaining link prediction task and regarding the $KL^-$ metrics are give in the Figure 6, 8 and 9 in the Appendix). DeepLIFT and GNNExplainer always, and Grad sometimes, fails to find the salient paths to explain the change, as they are designed for static graphs. In Appendix A.7, we provide cases where AxiomPath-Convex identifies edges and subgraphs that help make sense of the evolving predictions. In Appendix A.6, we analyze how long each component of the AxiomPath-Convex algorithms take on several datasets. In Appendix A.8, we analysis the limit of our method.

## 5 RELATED WORK

Differential geometry of probability distributions are explored in the field called "information geometry" Amari (2016), which has been applied to optimization Chen et al. (2020); Osawa et al. (2019); Kunstner et al. (2019); Seroussi & Zeitouni (2022); Soen & Sun (2021), machine learning Lebanon (2002); Karakida et al. (2020); Nock et al. (2017); Bernstein et al. (2020), and computer vision Shao et al. (2018). However, taking the geometric viewpoint of GNN evolution and its explanation is novel and has not been observed within the information geometry literature and explainable/interpretable machine learning.

Prior work explain GNN predictions on static graphs. There are methods explaining the predicted class distribution of graphs or nodes using mutual information Ying et al. (2019). Other works

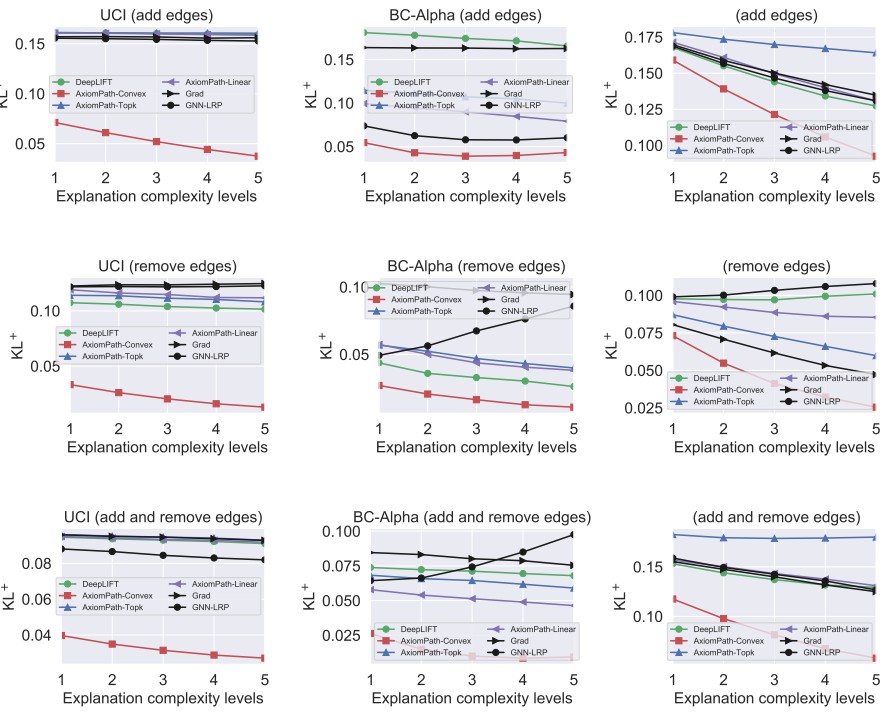

Figure 3: Average $KL^+$ on the link prediction and graph classification tasks. Each row is a dataset and each column is one evolution setting.

explained the logit or probability of a *single* class Schnake et al. (2020). CAM, GradCAM, GradInput, SmoothGrad, IntegratedGrad, Excitation Backpropagation, and attention models are evaluated in Sanchez-Lengeling et al. (2020); Pope et al. (2019) with the focus on explaining the static prediction of a single class. CAM and Grad-CAM are not applicable since they cannot explain node classification models Yuan et al. (2020b). DeepLIFT Shrikumar et al. (2017) and counterfactual explanations Lucic et al. (2021) do not explain multi-class distributions change over arbitrary graph evolution, as they assume $G_0$ is fixed at the empty graph. To compose an explanation, simple surrogate models Vu & Thai (2020) edges Schnake et al. (2020); Ying et al. (2019); Shrikumar et al. (2017); Lucic et al. (2021), subgraphs Yuan et al. (2021; 2020a) or graph samples Yuan et al. (2020a) , and nodes Pope et al. (2019) have been used to construct explanations. These works cannot axiomatically isolate contributions of paths that causally lead to the prediction changes on the computation graphs. Most of the prior work evaluates the faithfulness of the explanations of a static prediction. To explain distributional evolution, faithfulness should be evaluated based on approximation of the curve of evolution on the manifold so that the geometry will be respected. None prior work has taken a differential geometric viewpoint of distributional evolution of GNN. Optimally selecting salient elements to compose a simple and faithful explanation is less focused. With the novel reparameterization of curves on the manifold, we formulate a convex programming to select a curve that can concisely explain the distributional evolution while respecting the manifold geometry.

# 6 CONCLUSIONS

We studied the problem of explaining change in GNN predictions over evolving graphs. We addressed the issues of prior works that treat the evolution linearly. The proposed model view evolution of GNN output with respect to graph evolution as a smooth curve on a manifold of all class distributions. This viewpoint help formulate a convex optimization problem to select a small subset of paths to explain the distributional evolution on the manifold. Experiments showed the superiority of the proposed method over the state-of-the-art. In the future, we will explore more geometric properties of the construct manifold to enable a deeper understanding of GNN on evolving graphs.

## ACKNOWLEDGEMENTS

Sihong was supported in part by the National Science Foundation under NSF Grants IIS-1909879, CNS-1931042, IIS-2008155, and IIS-2145922. Yazheng Liu and Xi Zhang are supported by the Natural Science Foundation of China (No.61976026) and the 111 Project (B18008). Any opinions, findings, conclusions, or recommendations expressed in this document are those of the author(s) and should not be interpreted as the views of any U.S. Government.

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

## A  APPENDIX

You may include other additional sections here.

### A.1  MISC. PROOFS

Here we give the detailed derivations of Eq. (11).

$$D_{\text{KL}}(\Pr(Y|G_1)||\Pr(Y|G_0)) = \sum_{j=1}^{c} \Pr(Y = j|G_1) \log[\Pr(Y = j|G_1)/\Pr(Y = j|G_0)] \quad (13)$$

$$= \sum_{j=1}^{c} \Pr(Y = j|G_1)[z_j(G_1) - z_j(G_0)] - \log[Z(G_1)/Z(G_0)]$$

$$= \sum_{j=1}^{c} \Pr(Y = j|G_1)\mathbf{1}^\top \Delta C_{:j}(G_0, G_1) + \log Z(G_0) - \log \sum_{j=1}^{c} \exp\left\{ z_j(G^*) + \mathbf{1}^\top \underbrace{[C_{:j}(G_0) + \Delta C_{:j}(G_0, G_1)]}_{=C_{:j}(G_1)} \right\}$$

$$= \sum_{j=1}^{c} \Pr(Y = j|G_1)\mathbf{1}^\top \Delta C_{:j}(G_0, G_1) - \log Z(G_1) + \log \sum_{j=1}^{c} \exp\left\{ z_j(G^*) + \mathbf{1}^\top C_{:j}(G_0) \right\}$$

### A.2  SECOND-ORDERED APPROXIMATION OF THE KL DIVERGENCE WITH THE FISHER INFORMATION MATRIX

To help understand Eq. (9) that defines the Riemannian metric, we need to second-order approximation of the KL-divergence. We reproduce the derivations from the note "Information Geometry and Natural Gradients" posted on `https://www.nathanratliff.com/pedagogy/mathematics-for-intelligent-systems` by Nathan Ratliff (Disclaimer: we make no contribution to these derivations and the author of the note owns all credits). In the following, the term $\theta$ should be understood as the vector $\textbf{vec}(C_J(G_1))$ and $\delta$ should be understood as the difference vector $\textbf{vec}(\Delta C_J(G_1, G_0))$ in Eq. (9). $x$ is understood as the random variable $Y$, the class variable, in our case.

$$\text{KL}\left( p\left(x; \theta\right) \| p\left(x; \theta + \delta\right) \right)$$

$$\approx \int p\left(x; \theta\right) \log p\left(x; \theta\right) dx$$

$$- \int p\left(x; \theta\right) \left( \log p\left(x; \theta\right) + \left( \frac{\nabla_\theta p\left(x; \theta\right)}{p\left(x; \theta\right)} \right)^\top \delta + \frac{1}{2}\delta^\top \left( \nabla_\theta^2 \log p\left(x; \theta\right) \right) \delta \right) dx$$

$$= \underbrace{\int p\left(x; \theta\right) \log \frac{p\left(x; \theta\right)}{p\left(x; \theta\right)} dx}_{=0} - \underbrace{\left( \int \nabla_\theta p\left(x; \theta\right) dx \right)^\top \delta}_{=0}$$

$$- \frac{1}{2}\delta^\top \left( \int p\left(x; \theta\right) \nabla_\theta^2 \log p\left(x; \theta_t\right) \right) \delta$$

By assuming that the differentiation and integration in the second term can be exchanged, we have

$$\int \nabla p\left(x; \theta\right) dx = \nabla \int p\left(x; \theta\right) dx = \nabla 1 = 0$$

$$\nabla^2 \log p\left(x; \theta\right) = \frac{1}{p\left(x; \theta\right)} \nabla^2 p\left(x; \theta\right) - \nabla \log p\left(x; \theta\right) \nabla \log p\left(x; \theta\right)^\top$$

$$\mathrm{KL}\left(p\left(x;\theta\right)\|p\left(x;\theta+\delta\right)\right)$$

$$\approx -\frac{1}{2}\delta^{\top}\left(\int p\left(x;\theta\right)\nabla^2\log p\left(x;\theta\right)dx\right)\delta$$

$$= -\frac{1}{2}\delta^{\top}\underbrace{\left(\int \nabla^2 p\left(x;\theta\right)dx\right)}_{=0}\delta$$

$$+ \frac{1}{2}\delta^{\top}\underbrace{\left(\int p\left(x;\theta\right)\left[\nabla\log p\left(x;\theta\right)\nabla\log p\left(x;\theta\right)^{\top}\right]dx\right)}_{G(\theta_t)}\delta$$

The matrix $G\left(\theta\right)$ is known as the Fisher Information matrix.

### A.3 Optimize a curve on the link prediction task and graph classification task

Similar to node classification, according to the Eq.( 12), for the link prediction, we solve the following problem:

$$\min_{\substack{\vec{\mathbf{x}}(s)\in[0,1]^m \\ \vec{\mathbf{x}}'(s')\in[0,1]^{m'} \\ \|\vec{\mathbf{x}}(s)+\vec{\mathbf{x}}'(s')\|_1=n}} \mathbb{E}_{\mathsf{Pr}(Y|G_1)}[C_{:l}(G_1)-C_{:l}(\vec{\mathbf{x}}(s),\vec{\mathbf{x}}'(s'))]\boldsymbol{\theta}_{\mathsf{LP}}+\log\sum_{l=0}^{1}\exp\{z_j(G^*)+C_{:l}(\vec{\mathbf{x}}(s),\vec{\mathbf{x}}'(s'))\}\boldsymbol{\theta}_{\mathsf{LP}}$$

(14)

where $C_{:l}(G_1) = [\mathbf{1}^{\top}C_{:i}(G_1);\mathbf{1}^{\top}C_{:j}(G_1)]$, $C_{:l}(\vec{\mathbf{x}}(s),\vec{\mathbf{x}}'(s')) = [\vec{\mathbf{x}}(s)^{\top}(C_{:i}(G_0) + \Delta C_{:i}(G_0,G(s)));\vec{\mathbf{x}}'(s')^{\top}(C_{:j}(G_0) + \Delta C_{:j}(G_0,G(s')))]$.

For the graph classification, we solve the following problem:

$$\min_{\substack{\vec{\mathbf{x}}(s)\in[0,1]^m \\ \|\vec{\mathbf{x}}(s)\|_1=n}} \mathbb{E}_{\mathsf{Pr}(Y|G_1)}[C_{:g}(G_1) - C_{:g}(\vec{\mathbf{x}}(s)]\boldsymbol{\theta}_{\mathsf{GC}} + \log\sum_{j=1}^{c}\{\frac{\exp z_j(G^*)}{|\mathcal{V}|} + C_{:g}(\vec{\mathbf{x}}(s))\}\boldsymbol{\theta}_{\mathsf{GC}} \quad (15)$$

where $|\mathcal{V}|$ denotes the number of nodes in the graph, $C_{:g}(G_1) = \frac{\sum_{J\in\mathcal{V}}\mathbf{1}^{\top}C_{:j}(G_1)}{|\mathcal{V}|}$, $C_{:g}(\vec{\mathbf{x}}_{\mathbf{J}}(s)) = \frac{\sum_{J\in\mathcal{V}}\vec{\mathbf{x}}(s)^{\top}(C_{:j}(G_0)+\Delta C_{:j}(G_0,G(s))))}{|\mathcal{V}|}$.

### A.4 Attributing the change to paths

We describe the computation of $C_{p,j}$ in the previous section.

#### A.4.1 DeepLIFT for MLP

DeepLIFT Shrikumar et al. (2017) serves as a foundation. Let the activation of a neuron at layer $t + 1$ be $h^{(t+1)} \in \mathbb{R}$, which is computed by $h^{(t+1)} = f([h_1^{(t)}, \ldots, h_n^{(t)}])$, Given the *reference activation vector* $\mathbf{h}^{(t)}(0) = [h_1^{(t)}(0), \ldots, h_n^{(t)}(0)]$ at layer $t$ at time 0, we can calculate the *scalar reference activation* $h^{(t+1)}(0) = f(\mathbf{h}^{(t)}(0))$ at layer $t + 1$. The *difference-from-reference* is $\Delta h^{(t+1)} = h^{(t+1)} - h^{(t+1)}(0)$ and $\Delta h_i^{(t)} = h_i^{(t)} - h_i^{(t)}(0)$, $i = 1, \ldots, n$. With (or without) the 0 in parentheses indicate the reference (or the current) activations. The contribution of $\Delta h_i^{(t)}$ to $\Delta h^{(t+1)}$ is $C_{\Delta h_i^{(t)}\Delta h^{(t+1)}}$ such that $\sum_{i=1}^{n} C_{\Delta h_i^{(t)}\Delta h^{(t+1)}} = \Delta h^{(t+1)}$ (preservation of $\Delta h^{(t+1)}$).

The DeepLIFT method defines multiplier and the chain rule so that given the multipliers for each neuron to each immediate successor neuron, DeepLIFT can compute the multipliers for any neuron

to a given target neuron efficiently via backpropagation. DeepLIFT defines the multiplier as:

$$m_{\Delta h_i^{(t)} \Delta h^{(t+1)}} = C_{\Delta h_i^{(t)} \Delta h^{(t+1)}} / \Delta h_i^{(t)} \tag{16}$$

$$= \begin{cases} \theta_i^{(t)} & \text{linear layer} \\ \Delta h^{(t+1)} / \Delta h_i^{(t)} & \text{nonlinear activation} \end{cases}$$

If the neurons are connected by a linear layer, $C_{\Delta h_i^{(t)} \Delta h^{(t+1)}} = \Delta h_i^{(t)} \times \theta_i^{(t)}$ where $\theta_i^{(t)}$ is the element of the parameter matrix $\theta^{(t)}$ that multiplies the activation $h_i^{(t)}$ to contribute to $h^{(t+1)}$. For element-wise nonlinear activation functions, we adopt the *Rescale* rule to obtain the multiplier such that $C_{\Delta h_i^{(t)} \Delta h^{(t+1)}} = \Delta h^{(t+1)}$.

DeepLIFT defines the chain rule for the multipliers as:

$$m_{\Delta h_i^{(0)} \Delta h^{(T)}} = \sum_l \cdots \sum_j m_{\Delta h_i^{(0)} \Delta h_l^{(1)}} \ldots m_{\Delta h_j^{(T-1)} \Delta h^{(T)}} \tag{17}$$

### A.4.2 DEEPLIFT FOR GNN

We linearly attribute the change to paths by the linear rule and Rescale rule, even with nonlinear activation functions.

When $G_0 \to G_1$, there may be multiple added or removed edges, or both. These seemingly complicated and different situations can be reduced to the case with a single added edge. First, any altered path can only have multiple added edges or removed edges but not both. If there were a removed edge that is closer to the root than an added edge, the added edge would have appeared in a different path and the removed edge must be from an existing path leading to the root. If there were an added edge closer to the root than a removed edge, the nodes after the removed edge have no contribution in $G_0$ and the situation is the same as with added edges. Second, a path with removed edges only when $G_0 \to G_1$ can be treated as a path with added edges only when $G_1 \to G_0$. Lastly, as shown below, only the altered edge closest to the root is relevant even with multiple added edges. Let $U$ and $V$ be any adjacent nodes in a path, where $V$ is closer to the root $J$.

**Difference-from-reference of neuron activation and logits**. When handling the path with multiple added edges, we let the reference activations be computed by the GNN on the graph $G_0$, $G_{ref} = G_0$, the graph at the current moment is $G_1$, $G_{cur} = G_1$. For a path $p$ in $\Delta W_J(G_0, G_1)$, let $p[t]$ denote the node or the neurons of the node at layer $t$. For example, if $p = (I, \ldots, U, V, \ldots, J)$, $p[T]$ represents $J$ or the neurons of $J$, and $p[0]$ represents $I$ or the neurons of $I$. Given a path $p$, let $\bar{t} = \max\{\tau | \tau = 1, \ldots, T, p[\tau] = V \text{ and } p[\tau - 1] = U \text{ and if } (U, V) \text{ is newly added}\}$. When $t \geq \bar{t}$, the reference activation of $p[t]$ is $h_{p[t]}^{(t)}(G_0)$. While when $t < \bar{t}$, the reference activation of $p[t]$ is zero, because the message of $p[t]$ cannot be passed along the path $(p[t], \ldots, J)$ to $J$ in $G_0$, and the edge $(U, V)$ must be added to $G_0$ to connect $p[t]$ to $J$ in the path. We thus calculate the difference-from-reference of neurons at each layer for the specific path as follows:

$$\Delta h_{p[t]}^{(t)} = \begin{cases} h_{p[t]}^{(t)}(G_1) - h_{p[t]}^{(t)}(G_0) & t \geq \bar{t}, \\ h_{p[t]}^{(t)}(G_1) & \text{otherwise.} \end{cases} \tag{18}$$

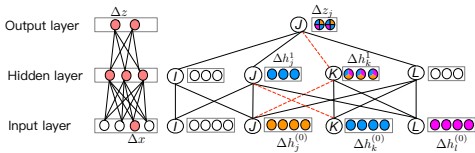

Figure 4: Circles in rectangles are neurons, and a neuron has a specific color if it contributes to the prediction change in a class. *Left*: DeepLIFT finds the contribution of an input neuron to the change in an output neuron of an MLP for link prediction, where the input layer is the output of a GNN. *Right*: A two-layer GNN. The four colored quadrants in $\Delta z_j$ at the top layer, which can be the input layer to the MLP, can be attributed to the changes in the input neurons at the input layer (e.g., the two blue quadrants at $J$ at the top is attributed to the blue neurons in node $K$ at the input layer through paths $(K, K, J)$ and $(K, J, J)$.

For example, in Figure 4, the added edge is $(J, K)$. For the path $p = (J, K, J)$ in $G_1$, $\bar{t} = 2$ and $\Delta h_j^{(0)} = h_j^{(0)}(G_1)$, because in $G_0$, the neuron $j$ at layer 0 cannot pass message to the neuron $j$ at the output layer along the path $(J, K, J)$ in $G_0$. The change in the logits $\Delta z_{p[t]}^{(t)}$ can be handled similarly.

**Multiplier of a neuron to its immediate successor.** We choose element-wise sum as the $f_{\text{AGG}}$ function to ensure that the attribution by DeepLIFT can preserve the total change in the logits such that $\Delta z_j = \sum_{p=1}^m C_{p,j}$. Then, $z_v^{(t)} = \sum_{U \in \mathcal{N}(V)} \left( \sum_{u \in U} h_u^{(t-1)} \theta_{u,v}^{(t)} \right)$, where $\theta_{u,v}^{(t)}$ denotes the element of the parameter matrix $\theta^{(t)}$ that links neuron $u$ to neuron $v$. According to the Eq. (16),

$$m_{\Delta h_u^{(t-1)} \Delta z_v^{(t)}} = \theta_{u,v}^{(t)}, \quad m_{\Delta z_v^{(t)} \Delta h_v^{(t)}} = \frac{\Delta h_v^{(t)}}{\Delta z_v^{(t)}}. \tag{19}$$

Then we can obtain the multiplier of the neuron $u$ to its immediate successor neuron $v$ according to Eq. (17):

$$m_{\Delta h_u^{(t-1)} \Delta h_v^{(t)}} = \frac{\Delta h_v^{(t)}}{\Delta z_v^{(t)}} \times \theta_{u,v}^{(t)}. \tag{20}$$

Note that the output of GNN model is $z_J$, thus $m_{\Delta h_{p[T-1]}^{(T-1)} \Delta z_j} = \theta_{p[T-1],j}^{(T)}$. We can obtain the multiplier of each neuron to its immediate successor in the path according to Eq. (20) by letting $t = T \to 1$.

After obtaining the $m_{\Delta h_{p[0]}^{(0)} \Delta h_{p[1]}^{(1)}}, \ldots, m_{\Delta h_{p[T-1]}^{(T-1)}, \Delta z_j}$, according to Eq. (17), we can obtain $m_{\Delta h_{p[0]}^{(0)} \Delta z_j}$ as

$$m_{\Delta h_{p[0]}^{(0)} \Delta z_j} = \sum_{p[0]} \cdots \sum_{p[T-1]} m_{\Delta h_{p[0]}^{(0)} \Delta h_{p[1]}^{(1)}} \cdots m_{\Delta h_{p[T-1]}^{(T-1)} \Delta z_j}. \tag{21}$$

**Calculate the contribution of each path.** For the path $p$ in $\Delta W_J(G_0, G_1)$, we obtain the contribution of the path by summing up the input neurons' contributions:

$$C_{p,j} = \sum_{p[0]} m_{\Delta h_{p[0]}^{(0)} z_j} \times \Delta h_{p[0]}^{(0)}, \tag{22}$$

where $p[0]$ indexes the neurons of the input (a leaf node in the computation graph of the GNN) and $\Delta h_{p[0]}^{(0)} = h_{p[0]}^{(0)}$.

---

**Algorithm 1** Compute $C_{p,j}$ for a target node $J$.

---

1: **Input**: two graph snapshots $G_0$ and $G_1$. Pre-trained GNN parameters $\theta_{GNN}$ for node classification,
2: Obtain the altered path set $\Delta W_J(G_0, G_1)$.
3: Initialize $C \in \mathbb{R}^{|W_J(G_0, G_1)| \times c}$ as an all-zero matrix
4: **for** $p \in \Delta W_J(G_0, G_1)$ **do**
5:     **if** $p$ contains removed edges **then**
6:         Reverse $G_0 \to G_1$ to $G_1 \to G_0$.
7:         Compute $C_{p,j}$ according to Eq. (22) and let $-C_{p,j}$ be the contribution of $p$ as $G_0 \to G_1$.
8:     **else**
9:         Compute $C_{p,j}$ according to Eq. (22) as the contribution of $p$ as $G_0 \to G_1$.
10:     **end if**
11: **end for**
12: **Output**: the contribution matrix $C$.

---

Algorithm 1 describes how to attribute the change in a root node $J$'s logit $\Delta z_j$ to each path $p \in \Delta W_J(G_0, G_1)$.

**The computational complexity of $C_{p,j}$.** Supposing that we have the $T$ layer GNN, the dimension of hidden vector $h_J^{(t)}$ of layer $t$ is $d_t (t = 1, 2, \ldots, T)$ and the dimension of the input feature vector is $d$. The time complexity of determining the contribution of each path is $\mathcal{O}(\prod_{t=1}^{T} d_t \times d)$. In the calculation, according to the Eq.20, we can obtain the multiplier $m_{\Delta h_u^{(t-1)} \Delta h_v^{(t)}}$. Then according to the Eq. 21 and the Eq. 22, we use the chain rule to obtain the final multiplier and then obtain the contribution. Because the multiplier $m_{\Delta h_u^{(t-1)} \Delta h_v^{(t)}}$ is sparse, obtaining the final multiplier matrix is also relatively fast. For the dense edge structure, the number of paths is large. But for these paths with the same nodes after layer $t$ (for example, the path $(K, K, J)$ and the path $(J, K, J)$), the multipliers after the layer $t$ are(for example $m_{\Delta h_k^{(1)} \Delta h_j^{(2)}}$) the same. The proportion of paths that can share multiplier matrices is large. Because of this, the calculation will speed up.

**Theorem 1.** *The GNN-LRP is a special case if the reference activation is set to the empty graph.*

*Proof.* Considering the path $p = (I, \ldots, U, V, \ldots J)$ on the graph $G_1$, we let $G_0$ is the empty graph. Then, $\Delta h_{p[t]}^{(t)} = h_{p[t]}^{(t)}(G_1), \Delta z_{p[t]}^{(t)} = z_{p[t]}^{(t)}(G_1), m_{\Delta h_u^{(t-1)} \Delta h_v^{(t)}} = \frac{\Delta h_v^{(t)}}{\Delta z_v^{(t)}} \times \theta_{u,v}^{(t)}$. While, for the GNN-LRP, when $\gamma = 0$, $R_j = z_j$, we note $\text{LRP}_{u,v}^{(t)} = \frac{h_u^{(t-1)} \theta_{u,v}^{(t)}}{\sum_{U \in N(V)} \sum_u h_u^{(t-1)} \theta_{u,v}^{(t)}} = \frac{h_u^{(t-1)} \theta_{u,v}^{(t)}}{z_v^{(t)}}$ that represents the allocation rule of neuron $v$ to its predecessor neuron $u$. The contribution of this path is

$$
\begin{aligned}
R_p &= \sum_i \cdots \sum_{p[T-1]} \text{LRP}_{i,p[1]}^{(1)} \cdots \text{LRP}_{p[T-1],j}^{(T)} R_j \\
&= \sum_i \cdots \sum_{p[T-1]} \frac{h_i^{(0)} \theta_{i,p[1]}^{(1)}}{z_{p[1]}^{(1)}} \cdots \frac{h_{p[T-1]}^{(T-1)} \theta_{p[T-1],j}^{(T)}}{z_j} z_j \\
&= \sum_i h_i^{(0)} \sum_{p[1]} \cdots \sum_{p[T-1]} \frac{h_{p[1]}^{(1)} \theta_{i,p[1]}^{(1)}}{z_{p[1]}^{(1)}} \cdots \theta_{p[T-1],j}^{(T)} \\
&= \sum_i m_{\Delta h_i^{(0)} \Delta z_j} h_i^{(0)} \\
&= C_{p,j}
\end{aligned}
$$

$\square$

## A.5 EXPERIMENTS

### A.5.1 DATASETS

- YelpChi, YelpNYC, and YelpZip Rayana & Akoglu (2015): each node represents a review, product, or user. If a user posts a review to a product, there are edges between the user and the review, and between the review and the product. The data sets are used for node classification.
- Pheme Zubiaga et al. (2017) and Weibo Ma et al. (2018): they are collected from Twitter and Weibo. A social event is represented as a trace of information propagation. Each event has a label, rumor or non-rumor. Consider the propagation tree of each event as a graph. The data sets are used for node classification.
- BC-OTC[3] and BC-Alpha[4]: is a who trusts-whom network of bitcoin users trading on the platform. The data sets are used for link prediction.
- UCI[5]: is an online community of students from the University of California, Irvine, where in the links of this social network indicate sent messages between users. The data sets are used for link prediction.
- MUTAG Morris et al. (2020): A molecule is represented as a graph of atoms where an edge represents two bounding atoms.

[3] http://snap.stanford.edu/data/soc-sign-bitcoin-otc.html
[4] http://snap.stanford.edu/data/soc-sign-bitcoin-alpha.html
[5] http://konect.cc/networks/opsahl-ucsocial

### A.5.2 EXPERIMENTAL SETUP

We trained the two layers GNN. We choose element-wise sum as the $f_{AGG}$ function. The logit for node $J$ is denoted by $z_J(G)$. For node classification, $z_J(G)$ is mapped to the class distribution through the softmax (number of classes $c > 2$) or sigmoid (number of classes $c = 2$) function. For the link prediction, we concatenate $z_I(G)$ and $z_J(G)$ as the input to a linear layer to obtain the logits. Then it be mapped to the probability that the edge $(I, J)$ exists using the sigmoid function. For the graph classification task, the average pooling of $z_J(G)$ of all nodes from $G$ can be used to obtain a single vector representation $z(G)$ of $G$ for classification. It can be mapped to the class probability distribution through the sigmoid or softmax function. We set the learning rate to 0.01, the dropout to 0.2 and the hidden size to 16 when we train the GNN model. The model is trained and then fixed during the prediction and explanation stages.

The node or edge or graph is selected as the target node or edge or graph, if $\mathrm{KL}(\mathrm{Pr}_J(G_1)\|\mathrm{Pr}_J(G_0)) > $ threshold, where threshold=0.001.

For the MUTAG dataset, we randomly add or delete five edges to obtain the $G_1$. For other datasets, we use the $t_{initial}$ and the $t_{end}$ to obtain a pair of graph snapshots. We get the graph containing all edges from $t_{initial}$ to $t_{end}$. Then two consecutive graph snapshots can be considered as $G_0$ and $G_1$. For Weibo and Pheme datasets, according to the time-stamps of the edges, for each event, we can divide the edges into three equal parts. On the YelpZip(both) and UCI, we convert time to weeks since 2004. On the BC-OTC and BC-Alpha datasets, we convert time to months since 2010. On other Yelp datasets, we convert time to months since 2004. See the table 2 for details.

Table 2: The details for datasets

| Datasets | Nodes | Edges | Settings | $t_{initial}$ | $t_{end}$ |
|---|---|---|---|---|---|
| **YelpChi** | 105,659 | 375,239 | **add (remove)** | 0 | [84, 90,96,102,108] |
| | | | **both** | [78,80,82,84] | [84,86,88,90] |
| **YelpNYC** | 520,200 | 1,956,408 | **add (remove)** | 0 | [78,80,82,84,86] |
| | | | **both** | [78,79,80,81] | [84,85,86,87] |
| **YelpZip** | 873,919 | 3,308,311 | **add (remove)** | 0 | [78,79,80,81,82] |
| | | | **both** | [338,340,342,344] | [354,356,358,360] |
| **BC-OTC** | 5,881 | 35,588 | **add (remove)** | 0 | [48,50,52,54,56] |
| | | | **both** | [24,26,28,30] | [48,50,52,54] |
| **BC-Alpha** | 3,777 | 24,173 | **add (remove)** | 0 | [48,51,54,57,60] |
| | | | **both** | [24,26,28,30] | [48,50,52,54] |
| **UCI** | 1,899 | 59,835 | **add (remove)** | 0 | [18,19,20,21,22] |
| | | | **both** | [16,17,18,19] | [19,20,21,22] |
| **Weibo** | 4,657 | | **add (remove)** | 0 | [1/3,2/3,1] |
| | | | **both** | [0,1/3] | [2/3,1] |
| **Pheme** | 5,748 | | **add (remove)** | 0 | [1/3,2/3,1] |
| | | | **both** | [0,1/3] | [2/3,1] |
| **MUTAG** | 17.93 | 19.79 | **add (remove, both)** | | |

To show that as $n$ increases, $\mathrm{Pr}_J(G_n)$ is gradually approaching $\mathrm{Pr}_J(G_1)$, we let n gradually increase. We choose $n$ according to the number of the altered paths . See the table 3 for details.

### A.5.3 QUANTITATIVE EVALUATION METRICS

We illustrate the calculation process of our method in Figure 5.

### A.5.4 EXPERIMENTAL RESULT

See the Figure 6 for the result on the $\mathrm{KL}^+$ on the YelpNYC, YelpZip and BC-OTC datasets. See the Figure 7, Figure 8 and Figure 9 for the result on the $\mathrm{KL}^-$ on the all datasets. The method AxiomPath-Convex is significantly better than the runner-up method.

Table 3: The the number of selected paths according to the number of altered paths.

| Task | settings | the number of altered paths | The the number of selected paths |
|---|---|---|---|
| Node classification | add, remove and both | $(1000, +\infty]$
$(500, 1000]$
$(100, 500]$
$(10, 100]$ | [15, 16,17,18,19]
[10,11,12,13,14]
[6,7,8,9,10]
[1,2,3,4,5] |
| Link prediction | add, remove and both | $(1000, +\infty]$
$(500, 1000]$
$(100, 500]$
$(10, 100]$ | [60,70,80,90,100]
[10,20,30,40,50]
[10,12,14,16,18]
[1,2,3,4,5] |
| Graph classification | add, remove and both | $(1000, +\infty]$
$(500, 1000]$
$(100, 500]$
$(10, 100]$ | [10,11,12,13,14]
[6,7,8,9,10]
[3,4,5,6,7]
[1,2,3,4,5] |

## A.6 SCALABILITY

**Running time overhead of convex optimization.** We plot the base running time for searching paths in $\Delta W(G_0, G_1)$ (or $\Delta W_I(G_0, G_1) \cup \Delta W_J(G_0, G_1)$) and attribution *vs.* the running time of the convex optimization. In Figure 10, we see that in the two top cases, the larger $\Delta W_J(G_0, G_1)$ (or $\Delta W_I(G_0, G_1) \cup \Delta W_J(G_0, G_1)$) lead to higher cost in the optimization step compared to path search and attribution. In the lower two cases, the graphs are less regular and the search and attribution can spend the majority computation time. The overall absolution running time is acceptable. In practice, one can design incremental path search for different graph topology, and more specific convex optimization algorithm to speed up the algorithm.

We plot the running of the baseline methods.(See the Figure 11,Figure 12 and Figure 13). The order of running time by the baseline methods is: AxiomPath-Convex, AxiomPath-Linear > DeepLIFT, AxiomPath-Topk > Gradient, GNNLRP. About for the GNNExplainer methods, they cost more time than AxiomPath-Convex when the the graph is small and they cost less time than DeepLIFT when the graph is large. Although the running time of Gradient and GNNLRP is less, the Gradient method cannot obtain the contribution value of the path, it only obtain the contribution value of the edge in the input layer. GNNLRP, like DeepLIFT, was originally designed to find the path contributions to the probability distribution in the static graph, and cannot handle changing graphs. If considering the running time of calculating path contribution values, we can use GNNLPR to obtain the paths contribution value in the $G_0$ and $G_1$ and subtracted them as the of the final contribution value. After obtaining $C_{p,j}$, we can still use our theory to choose the critical path to explain the change of probability distribution. GNNLPR can be a faster replacement for DeepLIFT.

## A.7 CASE STUDY

It is necessary to show that AxiomPath-Convex selects salient paths to provide insight regarding the relationship between the altered paths and the changed predictions.

On Cora, we add and/or remove edges randomly, and for the target nodes that the predicted class changed, we calculate the percentages of nodes on the paths selected by AxiomPath-Convex that have the same ground truth labels as the predicted classes on $G_0$ (class 0) and $G_1$ (class 1), respectively. We expect that there are more nodes of class 1 on the added paths, and more nodes of class 0 on the removed paths. We conducted 10 random experiments and calculate the means and standard deviations of the ratios. Figure 14 shows that the percentages behave as we expected. It further confirms that the fidelity metric aligns well with the dynamics of the class distribution that contributed to the prediction changes[6].

In Figure 15, on the MUTAG dataset, we demonstrate how the probability of the graph changes as some edges are added/removed. We add or remove edges, adding or destroying new rings in the molecule. The AxiomPath-Convex can identify the salient paths that justify the probability changes.

---

[6] AxiomPath-Convex has performance on Cora similar to those in Figure 2.

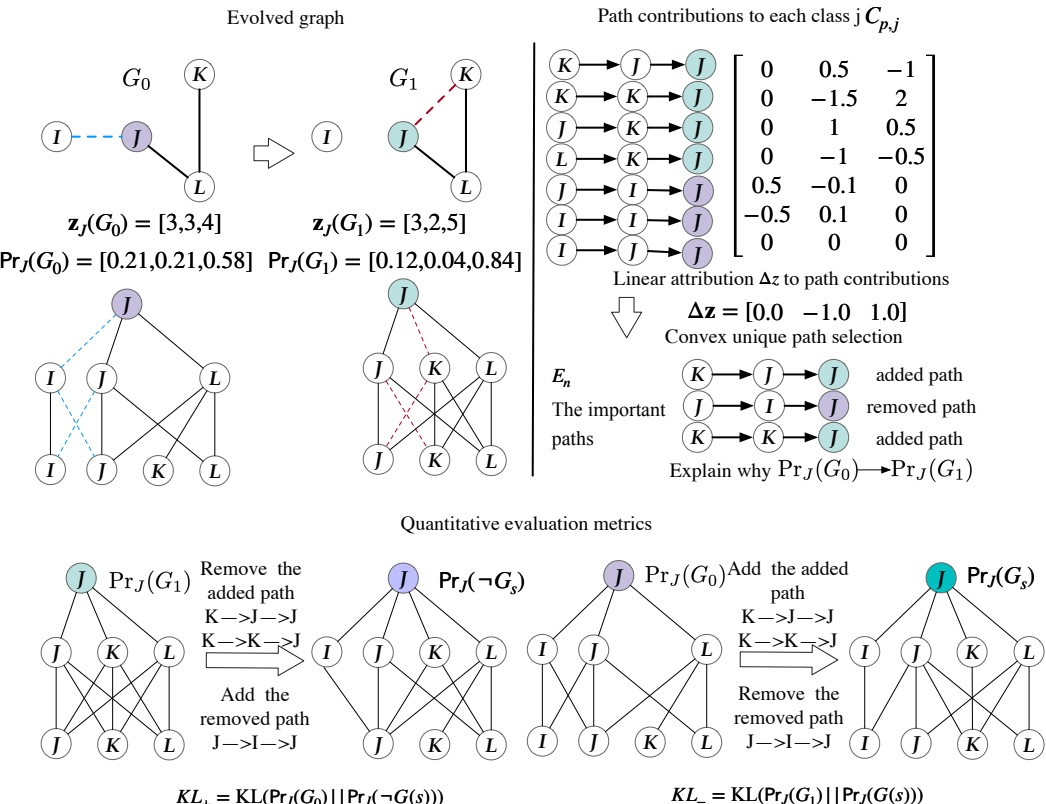

Figure 5: *Top left*: $G_0$ (e.g., a citation network) at time $t = 0$ is updated to $G_1$ at time $t = 1$ after the edge $(J, K)$ is added and the edge $(I, J)$ is removed, and the logits $\mathbf{z}_J(G_0)$ and predicted class distribution $\text{Pr}_J(G_0)$ of node $J$ changes accordingly. Prior counterfactual methods attribute the change to the edges $(J, K)$ and $(I, J)$. *Center left*: the GNN computational graph that propagates information from leaves to the root $J$. *Top right*: Any paths from the computational graph containing a dashed edge contribute to the prediction change, and we axiomatically attribute the logits changes to these paths with contribution $C_{p,j}$ (for the $p$-th path to the component $\Delta z_j$). *Center right*: Not all paths are significant contributors and we formulate a convex program to uniquely identify a few paths to maximally approximate the changes. *Bottom*: We show the calculation process of $\text{KL}^+$ and $\text{KL}^-$ after obtaining $E_n$. Other situations, including edge deletion, mixture of addition and deletion, and link prediction can be reduced to this simple case.

## A.8 FURTHER EXPERIMENTAL RESULTS

We analyzed how the method performs on the spectrum of varying $\text{KL}(\text{Pr}_J(G_1)||\text{Pr}_J G(0))$ for the YelpChi, YelpZip, UCI, BC-OTC and MUTAG datasets when edges are added and removed (See Figure 16). For some nodes with the lower $\text{KL}(\text{Pr}_J(G_1)||\text{Pr}_J G(0))$, the $\text{KL}^+$ or $\text{KL}^-$ is higher. Through the further analysis, we find that it may because of the $\text{Pr}_J(G_1)$ or $\text{Pr}_J(G_0)$ has all probability mass concentrated at one class. (See Figure 17). For the some target nodes/edges/graphs with the classification probability in $G_1$ or $G_0$ close to 1, the $\text{KL}^+$ or $\text{KL}^-$ is high. That means the selected paths may not explain the change of probability distribution well. When the classification probability is close to 1 in the $G_0(G_1)$, it is more difficult to select a few paths to make the probability distribution close $G_1(G_0)$, so the $\text{KL}^+$ or $\text{KL}^-$ is high.

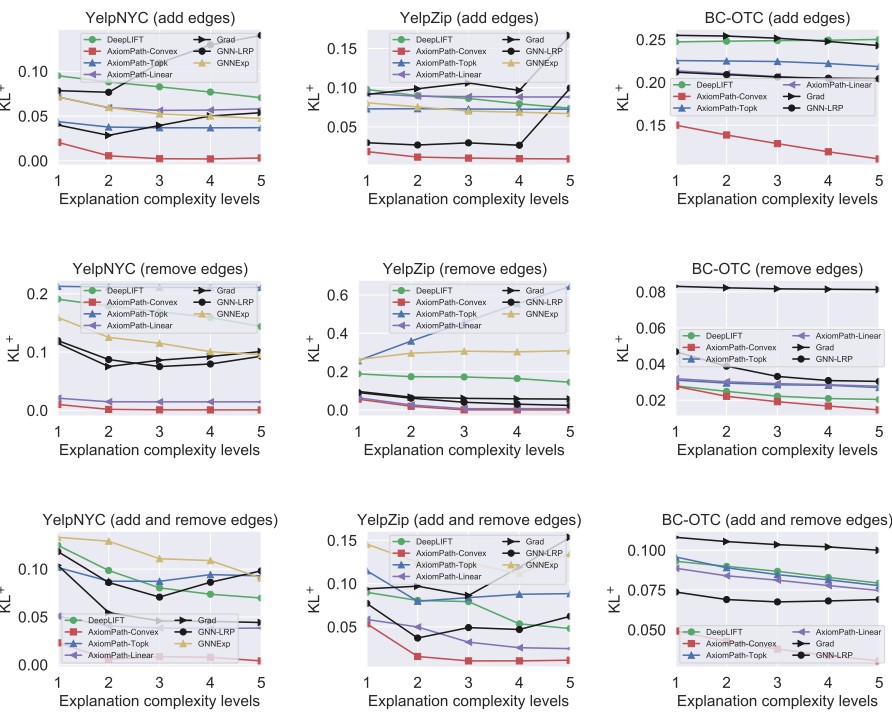

Figure 6: Performance on the node classification task and link prediction task. Each column is a dataset and each row is one setting. Each figure shows the $KL^+$ as $G_0 \rightarrow G_1$ for a pair of snapshots.

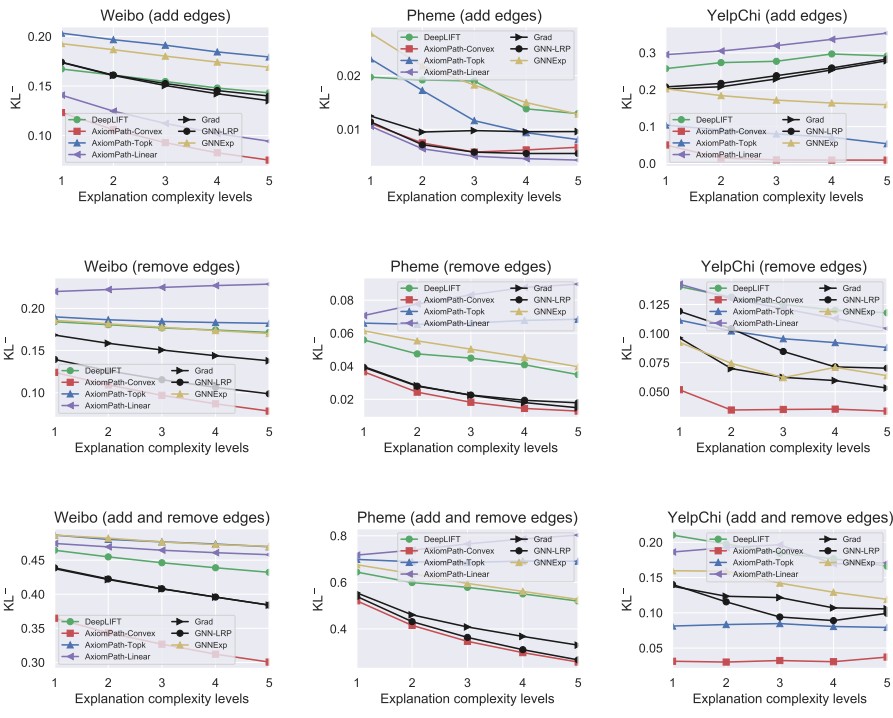

Figure 7: Performance on the node classification tasks. Each column is a dataset and each row is one setting. Each figure shows the $KL^-$ as $G_0 \rightarrow G_1$ for a pair of snapshots.

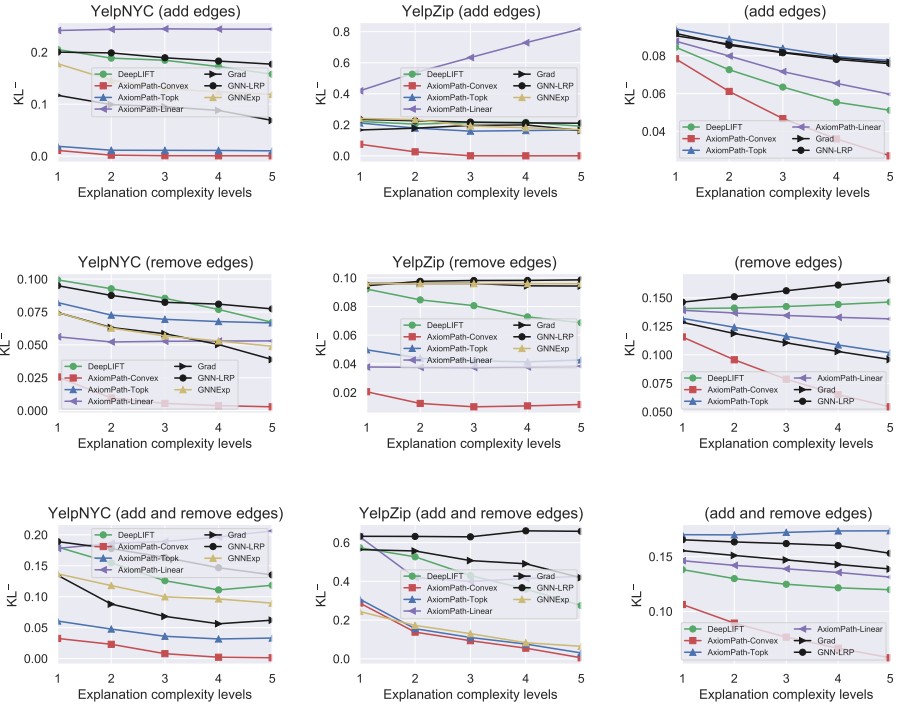

Figure 8: Performance on the node classification task and graph classification task. Each column is a dataset and each row is one setting. Each figure shows the KL$^-$ as $G_0 \rightarrow G_1$ for a pair of snapshots.

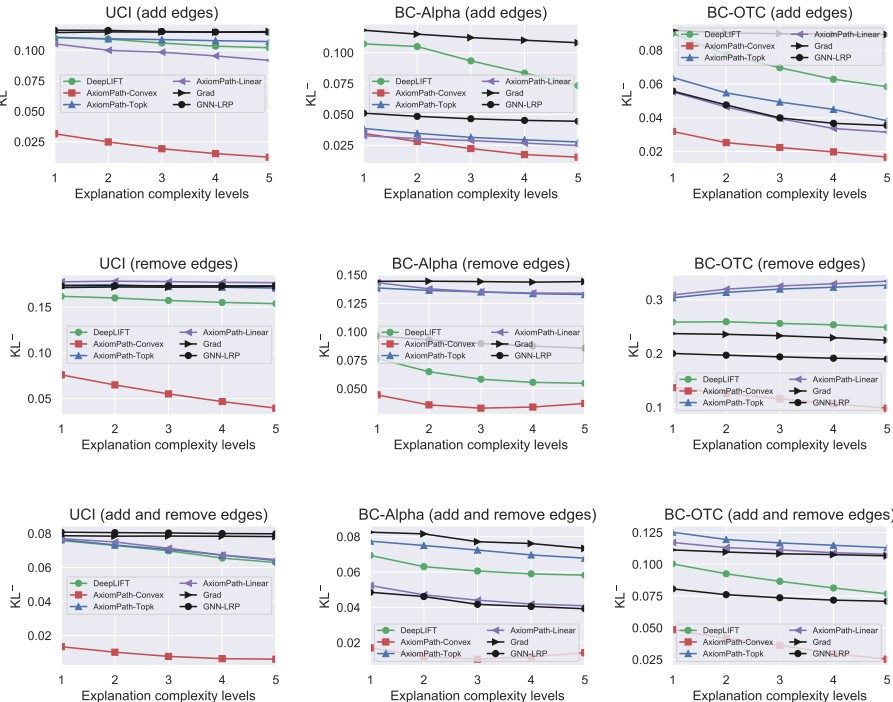

Figure 9: : Performance on the link prediction tasks. Average KL$^-$ on the link prediction and graph classification tasks. Each row is a dataset. Each column is one setting. Each figure shows the results of a pair of graph snapshots.

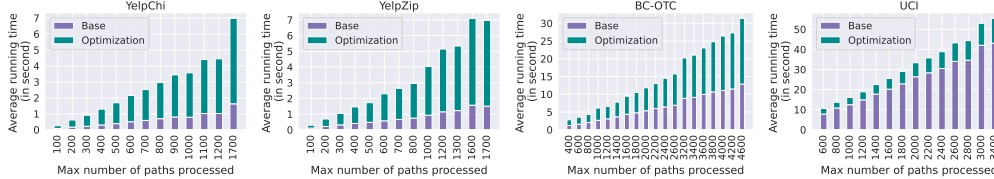

Figure 10: Decomposition of running time of AxiomPath-Convex.

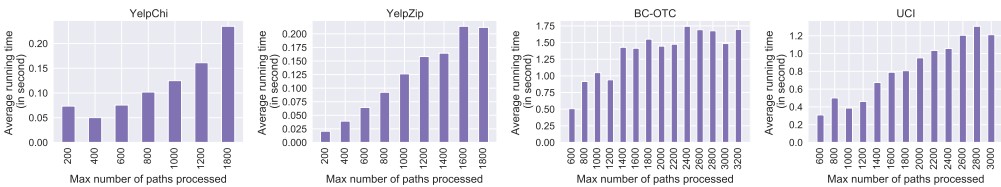

Figure 11: The running time of GNN-LRP.

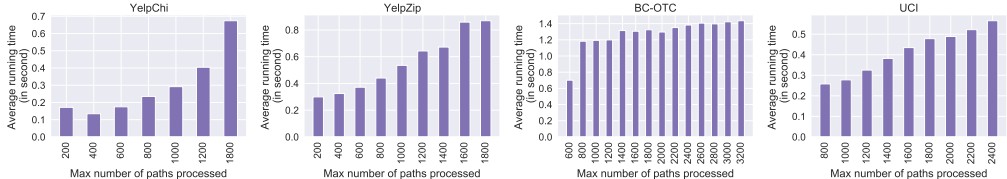

Figure 12: The running time of Gradient.

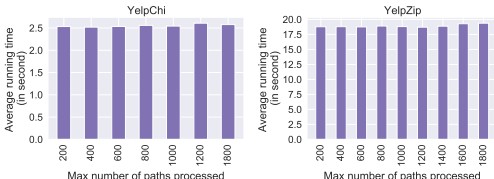

Figure 13: The running time of GNNExplainer.

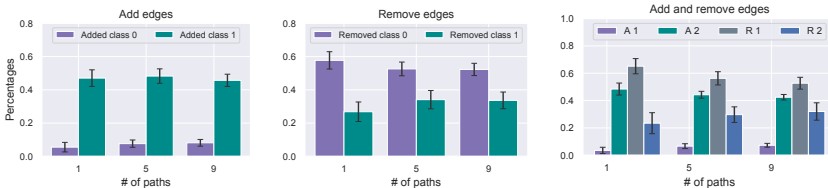

Figure 14: Case study on the Cora dataset. Adding edges only, removing edges only and both adding and removing edges. As AxiomPath-Convex selects different number of salient paths, we show the percentages of nodes on the selected paths from any previously predicted class on $G_0$ (class 0) and in any newly predicted class on $G_1$ (class 1).

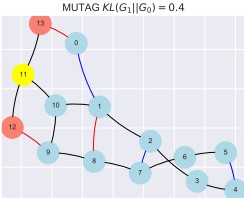
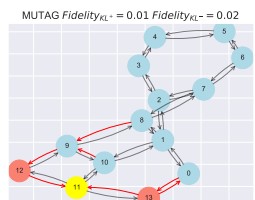

Figure 15: Case study on the MUTAG dataset. The circles denotes the nodes and the different types of nodes have different colors. Left: The black edges deonte the edges in the graph $G_0$ and $G_1$, the red/blue edges denote the added/removed edges. Right: The red edges on the paths selected by AxiomPath-Convex that lead to prediction changes. When adding or removing edges, the information gathered by neighbor nodes changed, thus affecting the classification probability. When masking or adding these red paths, $KL^+$ and $KL^-$ approach zero.

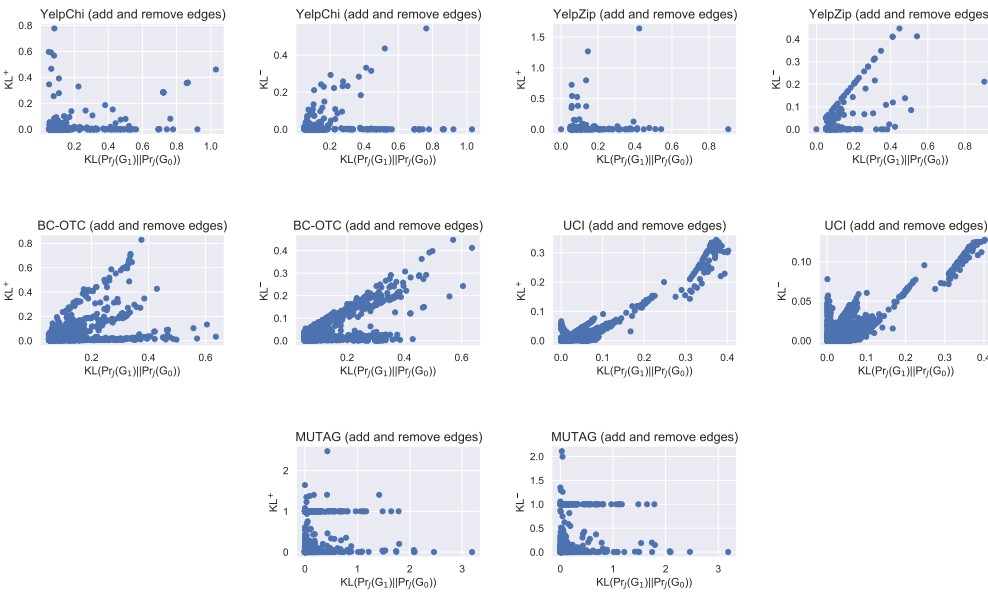

Figure 16: The $KL^-$ and $KL^+$ performance on the $KL(\text{Pr}_J(G_1)||\text{Pr}_J G(0))$.

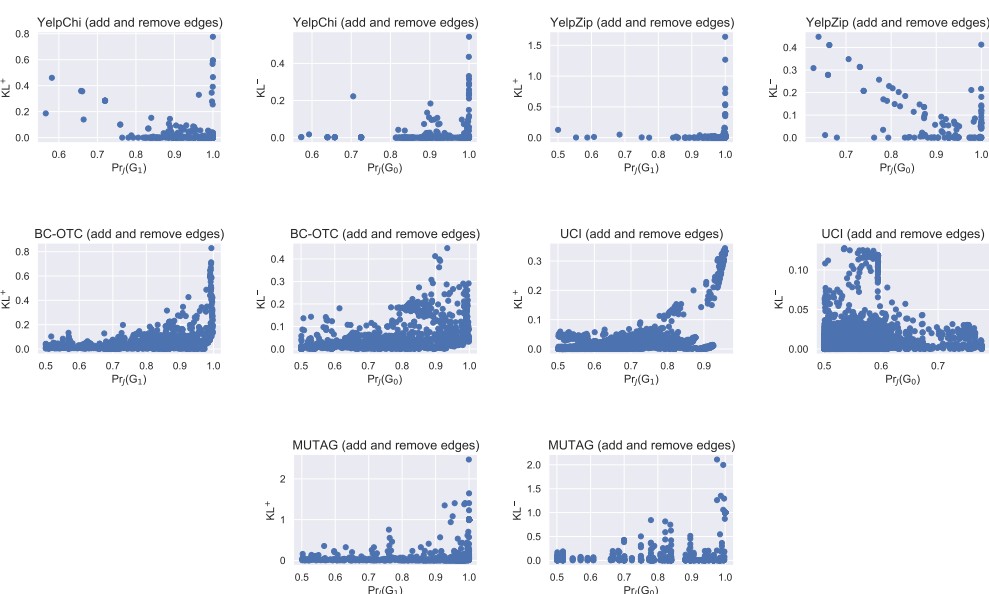

Figure 17: The $KL^-$ and $KL^+$ performance on the $Pr_J(G_1)$ or $Pr_J(G_0)$.

