# OpenReview forum: "A Differential Geometric View and Explainability of GNN on Evolving Graphs"
_ICLR.cc/2023/Conference — ICLR 2023 poster_

### Official Review · Reviewer_uccj · 2022-10-22

**Confidence:** 4
**Correctness:** 4
**Technical Novelty And Significance:** 3
**Empirical Novelty And Significance:** 2
**Recommendation:** 8

**Clarity, Quality, Novelty And Reproducibility:**

The writeup of the paper is very clear and the paper is easy to follow. Providing the complete hyperparameter detail will help assess the reproducibility. The empirical evidence clearly demonstrates that the approach is effective. Lastly, studying the problem through the lens of differential geometry is a new perspective.

**Strength And Weaknesses:**

1. The proposed method is intuitively clear and well supported by treferenes to the literature.
2. The results provided in the figures - 2, 3, 6, 7 and 8 clearly demonstrate that the proposed method outperforms the baselines (almost always).
3. Comparing the runtime against other baseline method/s (potentially the runner-up) will be helpful in identifying the runtime performance tradeoff.
4. As the work considers evolving graphs, discussing more than 2 snapshots in the evolution will be highly interesting. Have the authors performed some analyses on that? It will be interesting to study and helpful in demonstrating the effectiveness of the work.
5. Some analyses on the threshold described in section A.4.2 in order to select the target node/edge/graph will be useful. Studying how the method performs on the spectrum of varying $KL(P_J(G_1)||P_J(G_0))$ for the node/edge/graph $J$ can help explore the limit of the work.
6. Providing more details regarding the GNN hyperparameters and the training procedure can help assess the reproducibility of the work.


**Summary Of The Paper:**

This work studies the explainability on evolving graphs through the lens of “differential geometry”. While prior literature primarily focuses on static graphs, there are some works in the thread of evolving graphs against which the authors highlight certain differences in the introduction and related work. More concretely, the contributions of the work include - (i) embedding a manifold in an extrinsic euclidean space, (ii) designing extrinsic coordinates based on the contribution of paths to the predicted distributions on the computation graph of the GNN, (iii) defining a metric on the manifold and (iv) lastly, learning the curve that connects the two distributions and provide the desired explainability. The design of the coordinates is primarily based on prior work defining the set of $m$ length paths rooted at the node at hand $J$ with at least one altered edge from the initial graph to the final graph. The rationale is that these altered paths primarily lead to the altered predictions in the graph at the desired snapshot. The manifold is defined using the set $Pr(Y|G)$. The intrinsic dimension of the manifold is given by the sufficient statistics of $Pr(Y|G)$, which is parametrized by the contribution matrix (as described in eq 6, 7 and 8). Furthermore, they emphasize the idea that the evolution of $Pr(Y|G)$ is nonlinear on the manifold, which is based on the approximation of KL divergence provided in eq 9. This nonlinearity is attributed to the definitions - eq 6, 7 and 8. Lastly, they describe the curves using the specific types of parameterizations against the argument $\Delta{C}_{J}(G_0, G_1)$ in eq 10. The final objective described in eq 12 involves the constrained optimization of KL divergence term based on the curve formulation. Experiments have been performed on multiple benchmark datasets along with comparisons against the baseline explainability techniques.


**Summary Of The Review:**

Based on the questions and comments raised in the aforementioned sections, I lean towards acceptance of the work. I am willing to reconsider my score if the authors can try to address some of the questions.

---

> ### Author Response · Authors · 2022-11-19
> **Response to Reviewer uccj**
>
> Thank you for taking the time to review our paper!
>
> Q1: Comparing the runtime against other baseline method/s (potentially the runner-up) will be helpful in identifying the runtime performance tradeoff.
>
> A1: In Appendix A.6, Figures 10-13 of the new PDF, we report the runtime of the methods on several large datasets. The adopted DeepLIFT method for computing path contributions is not the fastest method that we know of. There are tricks to accelerate it. There is also another method called “GNNLRP”, that can compute path contributions much faster as it adopts the principle of the dynamic program by caching solutions to subproblems, without computing the contribution of each path independently.
>
> Q2: As the work considers evolving graphs, discussing more than 2 snapshots in the evolution will be highly interesting. Have the authors performed some analyses on that?
>
> A2: The proposed method can also handle more than 2 snapshots in the evolution. With a series of input graphs G_0, G_1, …G_t,…G_n, predicted Pr(Y|G) on any two graphs in the series can be compared and explained by our method. When deriving our formulation and methods, we referred to the two graphs G_0 and G_1, which should be interpreted more generally as any two graph snapshots in the series of graphs. There is no constraint preventing our framework from working on any two graph snapshots. In fact, our continuous parameterization of the manifold is more powerful and can model differences between any two computational graphs with a very subtle difference, even when the two input graphs are the same. The experiments are exactly conducted on naturally/artificially evolving graphs with more than two snapshots.
>
> Q3: Some analyses on the threshold described in section A.4.2 in order to select the target node/edge/graph will be useful. Studying how the method performs on the spectrum of varying KL(Pr_J (G_1)||Pr_J (G(0))) for the node/edge/graph can help explore the limit of the work.
>
> A3: in the original submission, we selected the target node/edge/graph based on a threshold of KL(Pr_J (G_1)||Pr_J (G(0))). The details are given in the second paragraph of Appendix A.5.2 in the new PDF. We added new experimental results and analyses based on a new threshold KL>=0.001 in Figures 16 and 17 in Appendix A.8 of the new PDF. The performance in KL+/KL- metrics depends on the topology of the graphs (e.g., high vs. low node degree), the original prediction Pr(Y|G), the KL between the original Pr(Y|G_0) and Pr(Y|G_1).
>
> Q4: Providing more details regarding the GNN hyperparameters and the training procedure can help assess the reproducibility of the work.
>
> A4: We describe the GNN architectures in the Section preliminaries, with more details in the first paragraph of Appendix A.5.2. Once the paper is accepted, we will open-source the codes so that others can reproduce it.

---

### Official Review · Reviewer_gpiW · 2022-10-22

**Confidence:** 3
**Correctness:** 4
**Technical Novelty And Significance:** 2
**Empirical Novelty And Significance:** 2
**Recommendation:** 6

**Clarity, Quality, Novelty And Reproducibility:**

The paper is clearly organized with interesting new ideas. A few places need more explanations.


**Strength And Weaknesses:**

Strength:
1. The main contribution is marginal: the proposed manifold-based distance metric for measuring the smoothing evolutions of a graph.
2. The experimental parts demonstrate that, for node classification tasks, the proposed method can find the most significant paths that explain the predictions of evolving graphs, thus, produce a good explanation.

Weakness:
1. Section 3.2 and 3.3 is kind of confusing. Need explanations about how Eq (10) and (11) are derived and calculated. How is the KL divergence second-ordered approximated by the Fisher information matrix?
2. The proposed methods select a subset of paths that contributes most to the expansibility, yet the number of selected paths is vague. Please explain the criteria for selecting the paths from Table 3.
3. A minor issue is the evolution of the graph is only constrained to adding or removing a subset of edges while the node features remain unchanged.


**Summary Of The Paper:**

This paper proposes a probabilistic metric for explaining the change of GNN predictions in an evolving graph from the differential geometric perspective. The author first reformulates the distribution of GNN predictions in the context of path contributions. Then, the author establishes the distance metric based on the approximated KL divergence between the distributions of each class for evolving graphs. And therefore, it constructs the Reiman Manifold for measuring distribution shifts on predictions regarding the change in graphs. Last, the authors formulate the problem of explaining GNN predictions into the problem of minimizing the KL divergence between distribution shifts. In experiments, a subset of paths, which contribute the most to the evolving process, is disabled, and the KL divergence is calculated and appears to be small, which validates the explanation quality of the proposed methods.

**Summary Of The Review:**

The paper is in general well written with coherence logic. The contributions are incremental. The experimental results appear to be good and support the conclusions. More explanations are needed to clarify a few ideas.

---

> ### Author Response · Authors · 2022-11-19
> **Response to Reviewer gpiW**
>
> Thank you for taking the time to review our paper!
>
> Q1: The experimental parts demonstrate that, for node classification tasks, the proposed method can find the most significant paths that explain the predictions of evolving graphs, thus, producing a good explanation.
>
> A1: The proposed geometric view is general and can be applied to various tasks of GNN with evolving graphs. At the bottom of Page 4, Eqs. (6)-(8) show that the view applies to node classification, link prediction, and graph classification tasks alike. In the experiment part, in Figures 2-3, and Figures 6-9 in the Appendix, we evaluate the method on node classification, link prediction, and graph classification tasks, showing that the method outperforms other baselines universally across these tasks and 8 datasets with both edge addition and removal.
>
> Q2: Section 3.2 and 3.3 is kind of confusing. Need explanations about how Eq (10) and (11) are derived and calculated. How is the KL divergence second-ordered approximated by the Fisher information matrix?
>
> A2: The detailed derivations of Eq (11) are in Appendix A.1. We clarify Eq. (10) in the updated PDF file: essentially, using the Kronecker product, we assign a time-dependent weight function x_p(s) for each path p for all c classes. For the KL divergence second-ordered approximated by the Fisher information matrix, we list the detailed steps of the proof in Appendix A.2.
>
> Q3: The proposed methods select a subset of paths that contributes most to the expansibility, yet the number of selected paths is vague. Please explain the criteria for selecting the paths from Table 3.
>
> A3: The number of selected paths depends on the number of edges involved in a prediction, and that depends on several factors, such as graph size, the number of edges in the neighborhood of a node or edge. The principle is to decide as a small number of paths as possible while the change in Pr(Y|G) can be approximated well. When many paths are involved in the prediction change (e.g., when a node is connected to many other neighbors), we need to involve relatively more paths to faithfully explain the change. We adjust the cut-offs of the number of selected paths in Table 3 based on that principle. From Figures 2 and 3, and 6-9 in the Appendix, we can see that the performances of all methods are stable at the selected cut-offs, while there is some diminishing return in the performance improvement as more paths are added/removed, indicating that more paths will not bring more faithfulness but to complicate the explanations unnecessarily.
>
> Q4: A minor issue is the evolution of the graph is only constrained to adding or removing a subset of edges while the node features remain unchanged.
>
> A4: We focus on the change in graph structure that leads to the probability distribution change, assuming that the node features remain the same. There is prior work focusing on explaining static GNN predictions as node feature vectors change. We intentionally study the problem of structural changes, as it is not only commonplace, but it also leads to the proposed differential geometric viewpoint and the manifold with a novel coordinate system based on path contributions on computational graphs. In contrast, the evolution of the node features is in the flat Euclidean space, which does not necessitate the novel parameterization that we proposed.
>
> Q5: The contributions are incremental.
>
> A5: We disagree that the contributions are incremental. We proposed a novel differential geometric viewpoint to GNN prediction on evolving graphs. Our work is complete in the sense that the differential geometric viewpoint applies to node classification, link prediction, and graph classification alike, with detailed derivations of the corresponding parameterization and experimental verification.
> Disclaimer: we are aware of the fact that differential geometry has been applied to study machine learning and computer vision before and we added references in the updated PDF.

---

> ### Author Response · Authors · 2022-12-06
> **Gentle reminder for feedback of our response**
>
> Dear Reviewer gpiW,
>
> We posted our official responses to your questions in the original review. May we humbly ask for your further feedback about our responses? We will be appreciative if we can have an opportunity to learn about your thinking and answer further questions.
>
> Thanks from the anonymous authors!

---

### Official Review · Reviewer_JnV5 · 2022-10-24

**Confidence:** 3
**Correctness:** 3
**Technical Novelty And Significance:** 3
**Empirical Novelty And Significance:** 3
**Recommendation:** 6

**Clarity, Quality, Novelty And Reproducibility:**

It is clearly well written with respect to the theory and algorithms behind the proposed method. The goals we want to achieve are common in recent research, but systematizing them in terms of graph evolution is new.


**Strength And Weaknesses:**

Strength

- Interpretation of the evolution of graphs is more difficult than, for example, changes in images, so it is worthwhile to realize them.
- It is organized according to a clearly systematized differential geometry scheme. Interpretations based on differential geometry are also used in the generative model, a method that is not new but is acceptable to those in this domain.

Weakness

- One concern is whether the graph can be handled continuously on the GNN manifold, since it will be discrete. That is, when perturbed on a GNN manifold, is there a graph corresponding to that point? Basically, all graphs may be considered as weighted graphs and considered continuous, which may be fine on a general level. However, I think the reference to this point is weak. The text may be structured in a misleading manner.
- This method constructs a large graph containing all data nodes. However, it is limited to the case where data including all nodes are present at training time, and it is likely that it will not work well when nodes that are not present at training time are input. This is considered a more cautionary situation than the usual Out of Distribution. It is not a big problem for cases where all the nodes are known or for the purpose of analyzing a model that has already been trained, so please clarify in what situations it can be used appropriately.
- I am not sure if what is shown in the experiment demonstrates the claim. I believe the authors' claim is that the proposed method makes graph evolution easier to understand. For example, it is expected to track the length of the corresponding paths and the evolution of the graph in multiple evolutions between the same starting and ending data, etc. Please clarify why the experiment demonstrates the claim of this paper.

**Summary Of The Paper:**

This paper proposes a method that gives a geometric viewpoint on the evolution of graphs and enhances their explanatory and interpretability. To capture changes in the graph of a discrete, we view the graph of each state as a subgraph of a larger graph, consider the space of probability distributions with respect to the links of that larger graph, and view them as curves on that space. The authors give specific procedures for interpreting the above perspectives based on differential geometry formulations. Experiments have also been conducted to verify the effectiveness of the method for the three tasks in the graph.

**Summary Of The Review:**

The evolution of a graph is difficult to understand as a graph as it is, and being able to understand it as a curve on a manifold is of great value. This method is clearly described in the framework of differential and information geometry. On the other hand, there are some unclear areas, such as necessary conditions that are implicit. Clarifying these would make for a better paper.

**Conclusion following discussion with the authors**

 It is now clear that the authors' claimed problem set is not affected by the concerns I had. On the other hand, I have the impression that the results obtained are more rudimentary than those discussed in similar directions, for example, in image representation learning. I think this is a good result for the future, so I have a positive impression, but I won't change the score.

---

> ### Author Response · Authors · 2022-11-07
> **Some clarification of the reviews from reviewer JnV5**
>
> Hi Reviewer JnV5,
>
> Thanks for your reviews. Before we respond to them more formally and thoroughly, we do have some clarification questions:
>
> 1. In "Interpretations based on differential geometry are also used in the generative model, a method that is not new but is acceptable to those in this domain", are there papers that we can refer to? A title would suffice. We are interested in making a reference if there is more related work.
> 2. In "On the other hand, there are some unclear areas, such as necessary conditions that are implicit.", we are not sure what "necessary conditions" in the submission are referred to.
>
> We are looking forward to your clarification to help us make more targeted responses!

---

> ### Author Response · Authors · 2022-11-19
> **Response to Reviewer JnV5**
>
> Thank you for taking the time to review our paper!
>
> Q1: Interpretations based on differential geometry are also used in the generative model.
>
> A1: thanks for pointing out the related papers, some of which we added as a reference. A generative model, or a neural network, is essentially a (smooth) function that creates a low-dimensional manifold in some high-dimensional extrinsic Euclidean space. We can interpret GNN similarly. However, our novelty is a new parameterization of the extrinsic Euclidean space based on path contributions, rather than elements in the input data (graphs).
>
> We aim to generate explanations of a computational process with respect to evolving input graphs, while we assume that the term “interpretation” in your review means “an overarching mathematical angle or framework to understand a model”.
>
> Q2: One concern is whether the graph can be handled continuously on the GNN manifold since it will be discrete. That is, when perturbed on a GNN manifold, is there a graph corresponding to that point?
>
> A2: Each point on the constructed manifold is a computational graph, with extrinsic coordinates being a set of path contributions on the corresponding computational graph to $Pr(Y|G)$. An input graph G can lead to multiple different computational graphs, which have neurons as nodes and dependencies between neurons as edges. You can imagine running different GNN models on the same input graph. On the other hand, a computational graph has sufficient information to determine a single input graph. Our contribution is to construct a layer of smooth representation between the observable evolution of Pr(Y|G) and the input graph G.
>
> We don’t parameterize the manifold using weights on edges of the input graph, which has been done in prior work, such as GNNExplainer. Our parameterization provides a more detailed explanation of the evolution in the computational graph.
>
> Q3: ... it is limited to the case where data including all nodes are present at training time, and it is likely that it will not work well when nodes that are not present at training time are input. This is considered a more cautionary situation than the usual Out of Distribution.
>
> A3: In the training time, we train a GNN model for a node classification, link prediction, or graph classification task. The GNN is fixed during test time when the input graph is evolving.
>
> For explaining the classification of a node J that is not present during the training time, if node J is unseen in G_0 but seen in G_1, we can find C_(p,j) (G) by setting the relevant activations at intermediate layers to zero on whichever graph that J is not present, and assuming that the logits at the output layer for node J are all zeros so that the probability distribution Pr(Y|G) is uniform. Given the found C_(p,j) (G), we can solve the convex optimization problem. The above handling of unseen node J is similar to the link prediction and the graph classification tasks.
>
> We are not concerned about prediction accuracy when generalizing a trained GNN to unseen data and Out-of-Distribution literature is only tangentially relevant. Instead, we care about the explanation quality (faithfulness and sparsity) during test time when the underlying input graph is evolving.
>
> Q4: I am not sure if what is shown in the experiment demonstrates the claim. I believe the authors' claim is that the proposed method makes graph evolution easier to understand.
>
> A4: For interpretability, the proposed method both concisely and faithfully explains the GNN distributional evolution. We focus on explanations based on the contributions of paths on the computational graphs of the GNN, not based on the input graph structures. That is, when the edges are added and/or removed, we want to know which paths caused the change in Pr(Y|G)
>
> Quantitatively, we define the evaluation metrics KL+ and KL- to show faithfulness. The set of path E_n contains the very few selected important paths which contribute the most to the change in the Pr(Y|G). We added a running example of these two metrics in Figure 5 in the new PDF file.
>
> Qualitatively, our convex optimization formulation and algorithm aim to find a small set of paths, making graph evolution easier for humans to digest and understand. Note: different levels of background knowledge of the users of the GNN models require different levels of detail in the explanations. Our method is aimed at users who are familiar with the GNN computing process, and they may be interested in questions, such as “what computational steps led to this wrong prediction when it was correct before this edge is added to the graph?” For the users with less knowledge, we visualize the influential components of the input graph based on path importance. See Figures 14 and 15.
>
> Q5: ... to track the length of the corresponding paths ...
>
> A5: the length of the curves on the manifold (we assume that’s what you refer to as “paths”) is not relevant to explaining the evolution of the prediction.

---

> > ### Comment · Reviewer_JnV5 · 2022-11-20
> > **Thanks for the clarification**
> >
> > Hi, Authors,
> >
> > Thanks for the clarification.
> >
> > I think I was somewhat misunderstood regarding your claim.
> > I thought the claim included pulling points on the manifold back to the actual (same level of input) graph, but now I understand that the main focus is on making the evolution of the graph graspable as a curve on the manifold.
> > From that perspective, the authors' rebuttal makes sense to me. In this case, I am interested in how the curves on the corresponding manifold actually differ in different evolutions with the same startpoint/endpoints, for example.
> >
> > Best,

---

> > > ### Author Response · Authors · 2022-11-21
> > > **Response to Reviewer JnV5**
> > >
> > > The construction of the manifold is independent of the start/end points of a curve, or of different curves between two fixed points. We assume a global reference point $Pr(Y|G^\ast)$ (which is a computational graph) on the manifold, which can be compared with any other point $Pr(Y|G)$.
> > >
> > > If the start/end points refer to two input graphs: assume that we have two series of graph snapshots for two different evolutions connecting the start/end input graphs, then the global reference point will put the computational graphs of all $Pr(Y|G)$ for any $G$ in the evolutions on the manifold with a global extrinsic coordinate. In this case, the two series of computational graphs are discrete points on the manifold, with intermediate points between any two adjacent points unknown. Our convex optimization problem is to find a unique curve that optimally approximates how one point approaches the other.
> > >
> > > If the start/end points refer to two places on the manifold: there can be infinitely many possible curves connecting these two points as functions $\gamma:[0,1]\to \mathcal{M}$. Therefore, there are different smooth curves representing different evolutions.
> > >
> > > In sum, different evolutions, while having the same start/end points, can be represented as different curves on the manifold.

---

> > > > ### Comment · Reviewer_JnV5 · 2022-11-21
> > > > **Response**
> > > >
> > > > I intend Startpoint/endpoint to be two points on the manifold. I understand, of course, that the paths are infinite. I would like to see some examples of that. I think it fits with intuition that the paths corresponding to the evolution of a simple graph and the paths corresponding to the evolution of a complex graph are more linear in the former.

---

> > > > > ### Author Response · Authors · 2022-11-23
> > > > > **Response to Reviewer JnV5**
> > > > >
> > > > > Comparing the curves on the manifold for different sorts of graph evolution can be a direction that needs more investigation, and we admit that this direction is not within the scope of this submission and can be part of the future work. Still, we can shed some light on it:
> > > > >
> > > > > A simpler graph will have a less number of altered paths on the computational graph than a more complex graph, thus will be on a lower-dimensional subspace than a more complex graph. However, the dimension of the curve is the same as the number of sufficient statistics of the exponential family. The increase in the number of extrinsic coordinates needs not necessarily increase the dimension of the curve. Furthermore, even a one-dimensional curve can be highly non-linear (e.g., the function $y=\sin(1/x)$ is smooth everywhere with higher and higher curvature as $x\to 0$).
> > > > >
> > > > > Figure 1 of the submission provides an example with two points on the manifold. One thing to note is that a point on the manifold does not necessarily correspond to an input graph, but to a computational graph. On the same manifold, we can equally place any other $\textnormal{Pr}(Y|G)$ computed by a different computational graph. Therefore, the manifold is independent of the start/end points that one has chosen (thanks to global extrinsic coordinates based on path contributions and the fictitious global reference point $\textnormal{Pr}(Y|G^\ast)$).

---

> ### Author Response · Authors · 2022-12-06
> **To Reviewer JnV5: want to be sure we address all your questions.**
>
> Dear Reviewer JnV5,
>
> We appreciate your engagement in a long conversation. If we have addressed all your concerns that negatively impact our submission, can we humbly ask if you can change the rating to reflect that? We are eager to answer more questions if you have more. In any case, we respect your honest rating and constructive feedback.
>
> Thanks from the anonymous authors!

---

### Official Review · Reviewer_yCBP · 2022-10-25

**Confidence:** 2
**Correctness:** 4
**Technical Novelty And Significance:** 2
**Empirical Novelty And Significance:** 2
**Recommendation:** 6

**Clarity, Quality, Novelty And Reproducibility:**

The work appears to be original in its method of constructing a smooth parameterization between probability distributions for GNNs. Several parts of the paper can be difficult to follow. For instance, the Riemannian metric on M is repeatedly emphasized, but the paper is somewhat vague about how this metric is invoked. On page 6, it is mentioned that "the curves should move according to the geometry of the manifold M(G, J)", but it's not made precise what this means. This is only elaborated on later in the section through the explanation "the above optimization does not change the Riemannian metric I(vec(CJ(G1))) at Pr(Y|G1) since Pr(Y|G1) is what we aim to approach on the manifold", but this is a trivial statement.

The experiments section of the paper could benefit from more detail on the setup. In particular, what GNN architectures were used, and what were your choices of hyperparameters?

**Strength And Weaknesses:**

Strengths:
-

- The evolution of paths is formulated nicely as a convex optimization problem with a theoretical motivation by KL divergence minimization.

Weaknesses:
-

- The focus is on interpolating between two graphs G0 and G1, but it's not clear why we care about intermediate stages when the data itself comes in discrete intervals, and the modifications to the graph are themselves discrete. The experiments do not seem to shed further light on this.

- The interpolation doesn't seem to use global time information, only the probability distributions P(Y|G0) and P(Y|G1) at consecutive time intervals.

- The convex optimization selects over a small number of paths in the computation graph which contribute the most to the change in the output vector. Determining the contribution of each path seems to be expensive to determine, especially for a GNN with many layers or a relatively dense edge structure. If this is not the case, it would be good to have some mention of the computational complexity of determining the highest-contributing paths.

**Summary Of The Paper:**

The paper introduces a new method of explaining how GNNs respond to evolutions of graph structure through adding or removing edges. It models the outputs of the GNN as a probability distribution depending on the input graph which varies over time. In contrast to previous work, time is taken as a continuous parameter. The most relevant edge additions and deletions are calculated and used in a convex optimization problem to determine a time-dependent smooth parameterization of the GNN.

**Summary Of The Review:**

The paper provides an interesting geometric perspective on time-dependent graph changes. But some of the choices of methods used seem arbitrary and could use more justification, specifically to how this particular choice of curve contributes to explainability or performance of the GNN.

---

> ### Author Response · Authors · 2022-12-06
> **Gentle reminder for feedback of our response**
>
> Dear Reviewer yCBP,
>
> We posted our official responses to your questions in the original review. May we humbly ask for your further feedback about our responses? We will be appreciative if we can have an opportunity to learn about your thinking and answer further questions.
>
> Thanks from the anonymous authors!

---

### Author Response · Authors · 2022-11-19
**Summary of response to reviewers and the update log**

Summary of responses:

To reviewers yCBP and JnV5:
1) Why use continuous evolution and manifold when input graph evolution is discrete: the manifold is about continuously changing of path contributions on the computational graph that computes Pr(Y|G), not about the discretely varying input graphs.
2) Are the evaluation metrics KL+ and KL- make sense: we respond to your questions below your reviews. We add a running example in Figure 5 of the Appendix to demonstrate why these two metrics make sense.

To reviewers uccj and yCBP: we provide detailed experimental settings in Appendix 5.2. We also explain that our method is not limited to just two snapshots of a graph, but can be applied to any two graph snapshots in a series of evolving graphs. We give detailed responses to your reviews.

To reviewer gpiW: Table 3 of Appendix 5.2 are justified. Please see the responses under your reviews.

Update logs (in the order of the paper sections/equations):
1) In the first paragraph of the Introduction, we made clear the setting of the paper where a sequence of graphs can be addressed.
2) We add a footnote on page 3 to clarify that Pr(Y|G(s)), which is later defined as a point in the manifold, does not necessarily correspond to concrete input graph G(s). The manifold is a collection of computational graphs, each of which has path contributions to Pr(Y|G(s)) as the extrinsic coordinates.
3) Define curves more formally at the end of Page 3.
4) Right before Section 3.2, we define the manifold M more clearly.
5) Above Eq. (9), we corrected typos in the KL divergence and explained the difference between the Riemannian and Euclidean metrics. Below Eq. (9), we added references to the derivation of the Fisher Information Matrix in Appendix A.2. We also introduce the Fisher information matrix at graph G_1 rather than G_0 for consistency with the discussion below Eq. (12). The math is the same.
6) Around Eq. (12), we corrected some typos. We updated Eq. (12) to make it more connected to Eq. (11). At the end of the paragraph, we explain how Eq. (12) respects the Riemannian metric.
7) Appendix A.1: add more details and correct typos in the derivation of Eq. (11). The texts around Eq. (11) are revised to make it more consistent.
8) Appendix A.2: add the derivation of the Fisher information matrix and the second-order approximation of the KL-divergence.
9) Appendix A.4.2: added the computational time complexity of C_{p,j}
10) Appendix A.5.2: added the GNN architectures in the first paragraph.
11) Appendix A.5.3: added a running example to explain the evaluation metrics KL+ and KL-.
12) Appendix A.6: added the running time of baseline methods.
13) Appendix A.8: added the relationship between the faithfulness performance and the original KL-divergence/Pr(Y|G_1)/Pr(Y|G_0).

---

### Decision · Program_Chairs · 2023-01-20

**Decision:**

Accept: poster

**Justification For Why Not Higher Score:**

The problem may not be of interest for a wide audience.

**Justification For Why Not Lower Score:**

The problem studied in the paper is interesting and the approach is novel and nice.

**Metareview: Summary, Strengths And Weaknesses:**

The authors propose a new model to explain the prediction of GNN on evolving graphs. More precisely the paper study the problem from a differential geometry perspective. The authors start by interpreting the GNN predictions as path contributions. Then, they use approximated KL divergence to establish a distance between the distributions of each class for evolving graphs. Finally, the authors reformulate the problem as the problem of minimizing the KL divergence between distribution shifts.

The paper presents new interesting idea and the experimental results are interesting. Although some part of the paper are a bit hard to read for non-expert in the field.

Overall, the paper would be a nice contribution to the ICLR program.

**Note From Pc:**

if the above contains the word "oral" or "spotlight" please see: "oral" presentation means -> notable-top-5% and "spotlight" means -> notable-top-25%. As stated in our emails, we are disassociating presentation type from AC recommendations